# Robust and Efficient Deep Hedging via Linearized Objective Neural Network

## Abstract

Deep hedging represents a cutting-edge approach to risk management for financial derivatives by leveraging the power of deep learning. However, existing methods often face challenges related to computational inefficiency, sensitivity to noisy data, and optimization complexity, limiting their practical applicability in dynamic and volatile markets. To address these limitations, we propose Deep Hedging with Linearized-objective Neural Network (DHLNN), a robust and generalizable framework that enhances the training procedure of deep learning models. By integrating a periodic fixed-gradient optimization method with linearized training dynamics, DHLNN stabilizes the training process, accelerates convergence, and improves robustness to noisy financial data. The framework incorporates trajectory-wide optimization and Black-Scholes Delta anchoring, ensuring alignment with established financial theory while maintaining flexibility to adapt to real-world market conditions. Extensive experiments on synthetic and real market data validate the effectiveness of DHLNN, demonstrating its ability to achieve faster convergence, improved stability, and superior hedging performance across diverse market scenarios.

## 1 Introduction

Deep hedging has revolutionized financial risk management in derivative markets by utilizing deep learning to address the complexities of modern financial environments. Traditional methods, such as the Black-Scholes framework (Black & Scholes, 1973), rely on closed-form solutions and simplifying assumptions, limiting their applicability in scenarios involving path dependence, transaction costs, or highly nonlinear relationships (Björk, 2009). By contrast, deep hedging incorporates data-driven techniques that can dynamically adapt to market conditions and model complex dependencies, offering a flexible framework for managing derivative liabilities (Buehler et al., 2019). This adaptability enables the framework to address key challenges, such as incorporating transaction costs (Kallsen & Muhle-Karbe, 2015), handling nonlinear payoff structures (Föllmer & Schied, 2011), and managing noisy market data (Karakida et al., 2019). Despite its potential, deep hedging faces critical barriers to practical implementation, including high computational costs, sensitivity to noisy data, and difficulties in seamlessly integrating deep learning models with traditional financial optimization techniques (Buehler et al., 2019; Hull et al., 2023; Shreve, 2005).

A significant challenge in deep hedging lies in the computational complexity of optimizing strategies across extended trajectories, especially in high-dimensional financial environments (Buehler et al., 2019; Hull et al., 2023). Traditional deep learning models often struggle with navigating non-convex optimization landscapes (Karakida et al., 2019), which can lead to unstable training dynamics and suboptimal solutions. These models also exhibit heightened sensitivity to noisy or incomplete market data (Gatheral, 2011; Shreve, 2005), making them vulnerable to overfitting and reducing their robustness in real-world scenarios. Moreover, a critical gap remains in aligning these methods with the stringent requirements of financial applications, including interpretability and regulatory compliance (Pesenti et al., 2024). Practical constraints, such as transaction costs and action-dependence, where hedging decisions must account for the cumulative impact of past trades, add another layer of complexity to the optimization process (Kallsen & Muhle-Karbe, 2015; Moresco et al., 2024). The dynamic nature of financial markets further intensifies these challenges,

requiring models to adjust to rapid changes in volatility, shifting asset correlations, and evolving liquidity conditions (Björk, 2009; Pesenti & Jaimungal, 2023).

These challenges underscore the need for solutions that are not only computationally efficient and robust but also adaptable to evolving market conditions. Financial markets are characterized by their inherent unpredictability, with extreme market events and out-of-distribution scenarios often testing the limits of traditional and machine learning-based approaches. To remain effective, hedging models must generalize beyond the conditions observed during training while maintaining stability and accuracy in their predictions. Furthermore, addressing transaction costs and market frictions requires methods that can dynamically optimize trading decisions without sacrificing cost efficiency. Achieving practical adoption also demands models that integrate seamlessly with well-established financial principles, ensuring interpretability and enabling compliance with regulatory standards. These multifaceted requirements necessitate the development of innovative frameworks capable of bridging theoretical advancements and practical applications, paving the way for more robust, scalable, and explainable deep hedging solutions.

To address these challenges, we propose a novel framework, Deep Hedging with Linearized-objective Neural Network (DHLNN), which transforms the training and optimization dynamics of deep learning-based hedging strategies. At its core, DHLNN introduces a nested optimization approach with periodic fixed-gradient updates, enabling the linearization of the objective function during inner iterations. This design simplifies the complex optimization landscape, reduces sensitivity to noisy financial data, and accelerates convergence. By decoupling gradient computation from parameter updates, DHLNN allows multiple optimization steps to be performed on a stable, linearized objective, significantly enhancing computational efficiency without compromising accuracy. By anchoring the hedging strategy in the Black-Scholes Delta, this integration not only enhances the model's robustness against market volatility but also provides a structured foundation for managing derivative liabilities effectively. Furthermore, the framework incorporates trajectory-wide optimization, capturing the holistic dynamics of the hedging process and minimizing residual risks over the entire underlying asset price trajectory over the lifetime of the derivatives. Extensive experiments on synthetic and real market data validate the effectiveness of DHLNN, demonstrating its ability to achieve faster convergence, improved stability, and superior hedging performance under a variety of market conditions. By addressing the challenges of computational efficiency and robust risk management, DHLNN represents a significant advancement in deep hedging, offering a practical and reliable solution for managing financial risks in dynamic and volatile markets.

The rest of this paper is structured as follows. Section 2 reviews the existing advancements in deep hedging and financial optimization, highlighting gaps addressed by our work. Section 3 explains the hedging context in the financial market and defines the hedging optimization problem. Section 4 details the proposed design of the DHLNN framework. Section 5 presents extensive experimental results, demonstrating the framework's performance under diverse market conditions and validating its robustness and efficiency. Finally, Section 6 concludes this work, and suggests promising directions for future research.

## 2 Related Work

The integration of machine learning into financial risk management, with a focus on hedging, has seen growing interest for its capacity to address practical complexities. Traditional approaches, such as the Black-Scholes framework (Black & Scholes, 1973), rely on closed-form solutions for derivative pricing and hedging. While theoretically sound, these methods struggle to accommodate market frictions, transaction costs, and path-dependent dynamics (Shreve, 2005). Deep hedging has revolutionized financial risk management in derivative markets by leveraging deep learning to navigate complex market dynamics and address challenges such as transaction costs and liquidity constraints (Buehler et al., 2019). Practical insights and applications of deep hedging, emphasizing its model-independent nature and ability to optimize hedging strategies, have also been highlighted in industry reports (Gao et al., 2023)(Neagu et al., 2024). Collectively, these works demonstrate the transformative potential of deep learning in improving risk management and hedging efficiency in financial markets.

Deep hedging frameworks often employ reinforcement learning (Cao et al., 2021) and neural network architectures (Ruf & Wang, 2019) to optimize hedging strategies in dynamic environments. These models

extend classical paradigms by incorporating market features and frictions, enabling adaptive risk management. However, challenges such as action dependence, non-convex optimization landscapes, and sensitivity to noisy data (Karakida et al., 2019) remain significant barriers to their practical deployment. Moreover, existing methods often require extensive computational resources, limiting their scalability.

The No-Transaction Band (NTB) Network (Imaki et al., 2021), which is a specialized method within the deep hedging framework, addresses the challenge of transaction costs by introducing a constrained trading strategy that limits activity to a no-transaction band. While this approach effectively reduces transaction costs, it also exhibits significant limitations. NTB's rigid constraints can restrict adaptability to rapidly changing market conditions and complex trading environments. Additionally, its design may limit the model's capacity to capture complex dependencies in market data, potentially reducing its efficacy in high-dimensional and volatile scenarios. These weaknesses highlight the need for more flexible and robust solutions to manage the multifaceted challenges of deep hedging in diverse market environments.

Recent advancements in the integration of financial theory with machine learning have also demonstrated promising directions for managing complex financial risks. Pesenti et al. (2024) propose a dynamic risk budgeting framework that employs dynamic risk measures to allocate risks across time and assets, providing a structured approach to risk management. Their use of an actor-critic method highlights the potential for machine learning to operationalize advanced financial concepts. Similarly, Moresco et al. (2024) develop a framework to model uncertainty in stochastic processes through robust risk measures and recursive representations, offering valuable tools for dynamic decision-making under uncertainty. Furthermore, Pesenti & Jaimungal (2023) extend these ideas to active portfolio management, employing a Wasserstein ball approach to balance benchmark tracking with distributionally robust optimization. While these approaches effectively incorporate financial theory into machine learning frameworks, they often require extensive computational resources to handle complex dynamics and adapt to evolving market conditions. Additionally, their focus on specific financial applications limits their generalizability to broader scenarios, such as hedging under high-dimensional, noisy, and volatile environments. These challenges underscore the need for more flexible, computationally efficient, and interpretable solutions that can seamlessly adapt to diverse market dynamics while maintaining robust performance.

In this work, we introduce DHLNN to address the persistent challenges of computational inefficiency, sensitivity to noisy data, and optimization complexity in deep learning-based financial applications. Unlike traditional approaches, DHLNN innovates by enhancing the training procedure itself. Through the integration of a periodic fixed-gradient optimization method and linearized training dynamics, our framework stabilizes training, accelerates convergence, and significantly improves robustness in volatile and dynamic market environments. The generalizability of this innovation sets DHLNN apart from other deep learning methods. By providing a streamlined, scalable solution that can be seamlessly integrated into diverse neural network models, our framework offers a universally applicable strategy for improving training efficiency and robustness. This positions DHLNN as a pivotal advancement not only for hedging and risk management but also for broader applications in financial machine learning. These contributions underline the potential of our approach to redefine how deep learning models are trained and applied in financial contexts.

## 3 Problem Formulation and Hedging Objectives

Hedging in derivative markets is a fundamental practice for mitigating financial risks associated with liabilities arising from derivative payoffs. These liabilities are directly tied to the price trajectory of the underlying asset, exposing traders to potential losses under unfavorable market conditions. The primary objective of hedging is to design optimal trading strategies that minimize the capital injection required to balance these liabilities while accounting for transaction costs and dynamic market conditions.

### 3.1 Hedging in Financial Markets

Hedging in financial markets addresses the challenge of mitigating risks associated with holding derivative contracts, which derive their value from the behavior of an underlying asset such as a stock (Schofield, 2021). For example, a European call option grants the holder the right to buy the stock at a predetermined strike

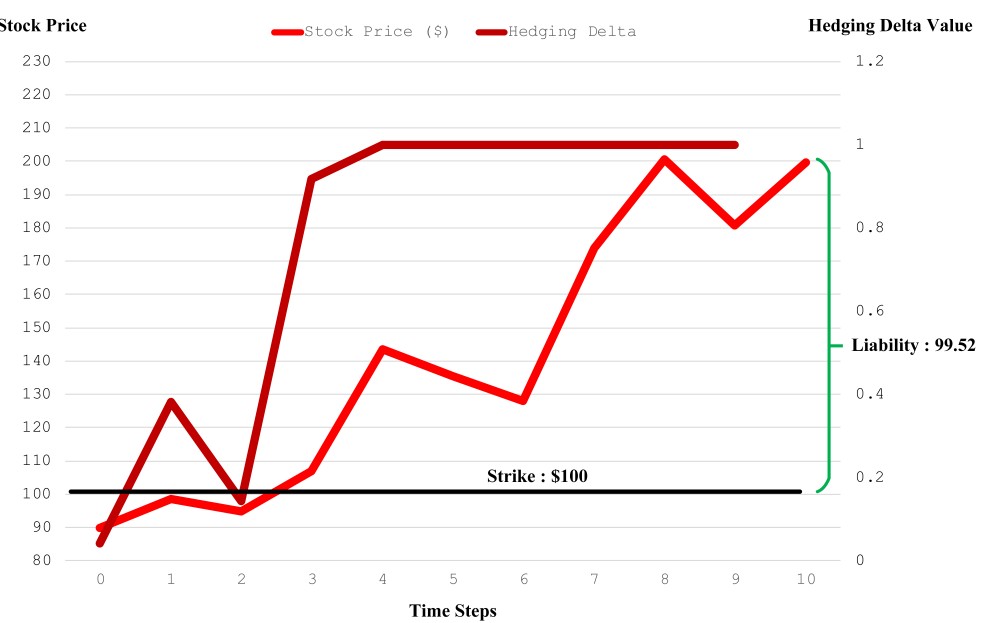

Figure 1: Illustration of the hedging procedure for a European call option. The option has a strike price of 100, a maturity of 10 days, and an initial stock price of 90, starting out-of-the-money. Using the Black-Scholes formula, the hedger dynamically adjusts the hedge delta at each step based on the stock price, time to maturity, and volatility (set at 0.2). The final stock price exceeds the strike price, creating a liability of approximately 99.52, which is offset to 11.81 through the hedging adjustments.

price on a future date, known as maturity (Black & Scholes, 1973). If the stock price at maturity exceeds the strike price, the option generates a profit for the option holder. However, for the entity that writes such derivatives, such as banks, market makers, hedge funds, or insurance companies, this scenario creates potential liabilities. These entities are obligated to fulfill the terms of the option contract, which may result in significant financial exposure when market movements are unfavorable. To mitigate these risks, option writers employ hedging strategies that dynamically adjust their positions in the underlying asset, aiming to offset potential losses and manage liabilities effectively (Hull et al., 2023).

To mitigate the risks associated with derivative liabilities, hedgers employ dynamic hedging strategies, which involve offsetting potential losses from derivative payoffs by trading the underlying asset. At any given moment, the hedger holds a position that represents the quantity of the underlying asset held. This position is dynamically adjusted based on changes in the asset's price and evolving market conditions to maintain a balance between the portfolio and the derivative liabilities (Björk, 2009). The central objective of this approach is to manage liabilities effectively by aligning trading actions with the derivative's sensitivities, ensuring that the portfolio adapts to market dynamics while minimizing residual risk.

Several critical factors influence the dynamics of hedging strategies. Log moneyness, defined as $\log\left(\frac{P_{t_i}}{P_s}\right)$, provides a standardized measure of the relationship between the underlying asset price $P_{t_i}$ and the derivative's strike price $P_s$ (Gatheral, 2011). This metric is crucial for understanding whether the derivative is in-the-money, at-the-money, or out-of-the-money and serves as a key input for pricing models and hedging decisions (Black & Scholes, 1973). Time to expiration, expressed as $T - t_i$, significantly affects the derivative's value by quantifying the remaining duration until maturity and reflecting the time decay effect (Björk, 2009). Volatility, $\sigma$, captures market expectations of price fluctuations, directly quantifying risk and shaping the effectiveness of the hedge (Shreve, 2005; Hull et al., 2023). By incorporating these factors, hedgers can refine their strategies to achieve precise and cost-effective adjustments to the holding of the underlying asset, making a trade-off between liability and trading efficiency.

The primary objective of hedging is risk management, not profit maximization. A hedger seeks to maintain a self-financed portfolio where trading gains or losses from the underlying asset effectively offset the liabilities arising from the derivative contract. This self-financed approach ensures that all capital adjustments occur

within the portfolio itself, avoiding the need for external funding except to cover any residual liabilities. The effectiveness of a hedging strategy is evaluated using Profit and Loss (PNL), which captures the financial outcome by balancing trading gains against derivative liabilities (Föllmer & Schied, 2011). An optimal hedging strategy aims for a neutral PNL, indicating that liabilities are offset with minimal residual risk (Hull et al., 2023). Deviations in PNL, whether positive or negative, highlight inefficiencies in the strategy, such as over-hedging or under-hedging, which can compromise its effectiveness (Shreve, 2005).

An illustration of the hedging procedure is presented in Fig. 1, using the example of a European call option with a strike price of 100 and a maturity of 10 days. Initially, the stock price is 90, placing the option out-of-the-money since the stock price is below the strike price. The option writer, responsible for delivering the payoff if the option is exercised, employs a dynamic hedging strategy to manage the liability risk at maturity. At each time step, the hedge delta, calculated using the Black-Scholes formula, is adjusted based on the current stock price, time remaining until maturity, and market volatility (set at 0.2). These adjustments align the portfolio with the evolving liability from the option's payoff. By maturity, the stock price has risen above the strike price, resulting in a liability of approximately 99.52. However, the dynamic hedging strategy significantly reduces this liability, resulting in a final Profit and Loss (PNL) of $-11.81$, effectively mitigating most of the financial risk.

The absence of a final hedging adjustment at maturity is purposefully designed to illustrate the impact of dynamic hedging along the entire trajectory. This approach emphasizes the necessity of continuous hedging efforts in mitigating liability risks before the option's maturity. Without introducing transaction costs in this simplified scenario, the results clearly demonstrate the effectiveness of this fundamental hedging strategy in reducing exposure and minimizing risk over time, highlighting the critical role of intermediate hedging actions.

### 3.2 Hedging Liabilities and Transaction Costs

Liabilities associated with derivative contracts depend on their payoff structures. For a European call option, the liability at maturity is given by

$$Z = \max(P_T - P_s, 0), \tag{1}$$

where $P_T$ is the terminal price of the underlying asset. In contrast, the liability of a Lookback call option, a path-dependent derivative, is expressed as

$$Z = \max\left(\max\{P_{t_i}\}_{0 \leq t_i \leq T} - P_s, 0\right), \tag{2}$$

where the payoff depends on the maximum price achieved by the underlying asset during the contract's duration. This path dependency introduces additional complexity in pricing and hedging (Shreve et al., 2004).

To accurately represent the value of the underlying asset, the Weighted Average Price (WAP) is utilized. Constructed from the order book, the WAP series is defined as

$$P = \{P_{t_i} \mid P_{t_i} > 0\}_{0 \leq t_i \leq T}, \tag{3}$$

where $P_{t_i}$ denotes the price at time $t_i$ within the observation period $[0, T]$. The WAP incorporates both liquidity and market depth by averaging bid and ask prices, weighted by their respective sizes (Gould et al., 2013). This approach ensures that price calculations account for the available volume at each level of the order book, providing a more nuanced and accurate measure. WAP is particularly popular in modern financial markets, offering a liquidity-adjusted metric that supports informed trading decisions. For a more detailed explanation of WAP, please refer to Appendix A.

Transaction costs, an integral aspect of hedging, introduce realistic market frictions. These costs are modeled as proportional to the traded value, with the cumulative transaction costs defined as

$$C_{t_{i+1}} = C_{t_i} + cP_{t_i}|\delta_{t_{i+1}} - \delta_{t_i}|, \tag{4}$$

where $\delta_{t_i}$ represents the hedge ratio at time $t_i$, indicating the proportion of the underlying asset held in the portfolio, and $c$ is the transaction cost rate. By penalizing frequent trading, transaction costs encourage efficient strategies that balance risk mitigation and economic feasibility (Gârleanu & Pedersen, 2013).

### 3.3 Optimization Formulation for Hedging

The primary objective of the hedging strategy is to minimize the capital injection $P_L$ required to meet liabilities while adhering to the self-financing condition

$$P_L - Z + \sum_{t_i=0}^{T-1} \delta_{t_i} \Delta P_{t_i} - c \sum_{t_i=0}^{T-1} |\delta_{t_{i+1}} - \delta_{t_i}| P_{t_i} = 0, \qquad (5)$$

where $\Delta P_{t_i} = P_{t_{i+1}} - P_{t_i}$ represents the price change of the underlying asset, and $\delta_{t_i}$ denotes the hedge ratio at time $t_i$. This condition ensures that all trading activities are self-financed, meaning the hedger's portfolio adjustments are funded by the portfolio itself without external capital inflows, except for the initial injection $P_L$.

To quantify the effectiveness of the hedging strategy, we adopt a framework based on indifference pricing. The indifference price, denoted by $q(Z)$, represents the additional capital that must be injected into the portfolio to make the hedger indifferent between having the liability $Z$ and not having any liability, under an optimal hedging strategy. This concept bridges the theoretical and practical aspects of hedging by evaluating the trade-off between minimizing risk and accounting for market frictions such as transaction costs (Föllmer & Schied, 2011).

The convex risk measure $\rho$ is a key component of this framework. It is a monotonic and cash-invariant function that quantifies the residual risk of the portfolio after hedging (Ben-Tal & Teboulle, 2007). By minimizing $\rho$, we ensure that the portfolio is optimally adjusted to mitigate risk while controlling trading costs. The optimization problem for determining the hedging strategy is expressed as

$$\min_{\delta} \rho \left( -Z + \sum_{t_i=0}^{T-1} \delta_{t_i} \Delta P_{t_i} - c \sum_{t_i=0}^{T-1} |\delta_{t_{i+1}} - \delta_{t_i}| P_{t_i} \right), \qquad (6)$$

where $-Z$ represents the liability, $\sum_{t_i=0}^{T-1} \delta_{t_i} \Delta P_{t_i}$ accounts for the cumulative change in the portfolio value arising from trading the underlying asset, and the term involving $c$ penalizes excessive trading activity.

The indifference price $q(Z)$ is then defined to capture the minimal capital required to neutralize the liability $Z$ while reflecting the costs and risks of hedging, which is expressed as

$$q(Z) = \inf_{\delta} \rho \left( -Z + \sum_{t_i=0}^{T-1} \delta_{t_i} \Delta P_{t_i} - c \sum_{t_i=0}^{T-1} |\delta_{t_{i+1}} - \delta_{t_i}| P_{t_i} \right) - \inf_{\delta} \rho \left( \sum_{t_i=0}^{T-1} \delta_{t_i} \Delta P_{t_i} - c \sum_{t_i=0}^{T-1} |\delta_{t_{i+1}} - \delta_{t_i}| P_{t_i} \right), \quad (7)$$

where the first term quantifies the minimal residual risk when the hedger has the liability $Z$, while the second term represents the minimal residual risk in the absence of liability. The difference between these two terms, $q(Z)$, provides a direct measure of the financial burden imposed by the liability $Z$. Essentially, $q(Z)$ corresponds to the additional cash injection required to ensure that the portfolio's risk profile, after optimal hedging, is equivalent to that of a portfolio without any liabilities (Artzner et al., 1999). More detailed explanations of the indifference price $q(Z)$ and its derivation are provided in Appendix B.

## 4 Deep Hedging with Nested Training Procedure via Linearized Objective

This section introduces a robust framework for dynamic hedging optimization that addresses key challenges in financial risk management and computational efficiency. At the heart of this framework is the linearized objective, which transforms traditional backpropagation into a nested training procedure. By constructing a local linear approximation of neural network outputs at each iteration, the framework simplifies the optimization process. A key innovation is the periodic fixed-gradient approach during inner iterations, enabling efficient parameter updates on a linearized target. This reduces sensitivity to noisy data, accelerates convergence, and enhances training stability, making the framework well-suited to high-volatility financial markets.

The nested optimization process alternates between outer iterations, where gradients are computed and fixed, and inner iterations, which optimize the linearized objective. This design improves interpretability, reduces computational overhead, and ensures robust performance under challenging market conditions. To further enhance practicality, the Black-Scholes Delta is integrated as an anchor, ensuring that neural network outputs remain consistent with well-established financial theories. The proposed framework demonstrates significant advantages in robustness, efficiency, and interpretability. It dynamically adjusts hedging positions while minimizing trading costs, ensuring adaptability across diverse market conditions.

### 4.1 Linearized-objective Neural Network Framework for Nested Optimization

The proposed framework leverages the gradient information of the neural network to construct a linear approximation of the objective function, combining nested optimization with linearized training dynamics, the approach addresses key challenges in financial markets, such as noisy data, computational inefficiencies, and non-stationarity.

At the core of the framework is the representation of the hedging position $\delta_{t_i}$ as the neural network output $f(\boldsymbol{x}_{t_i}, \boldsymbol{w})$, where $\boldsymbol{x}_{t_i}$ contains market features such as log moneyness, time to maturity, and volatility, and $\boldsymbol{w}$ represents the neural network parameters. The PNL value denoted by $V$ is formulated as

$$V = -Z + \sum_{t_i=0}^{T-1} f(\boldsymbol{x}_{t_i}, \boldsymbol{w})\Delta P_{t_i} - c\sum_{t_i=0}^{T-1} \left|f(\boldsymbol{x}_{t_{i+1}}, \boldsymbol{w}) - f(\boldsymbol{x}_{t_i}, \boldsymbol{w})\right|P_{t_i}, \tag{8}$$

where $Z$ is the liability, $\Delta P_{t_i} = P_{t_{i+1}} - P_{t_i}$ is the price change of the underlying asset, and $c$ represents transaction costs.

To streamline optimization, the neural network output is approximated by a linear function around the current parameter state $\boldsymbol{w^r}$ as

$$\hat{f}(\boldsymbol{x}_{t_i}, \boldsymbol{w}) = f(\boldsymbol{x}_{t_i}, \boldsymbol{w^r}) + \nabla_{\boldsymbol{w}} f(\boldsymbol{x}_{t_i}, \boldsymbol{w^r})^\top (\boldsymbol{w} - \boldsymbol{w^r}), \tag{9}$$

where $f(\boldsymbol{x}_{t_i}, \boldsymbol{w^r})$ is the output at $\boldsymbol{w^r}$, and $\nabla_{\boldsymbol{w}} f(\boldsymbol{x}_{t_i}, \boldsymbol{w^r})$ is the gradient of the output with respect to $\boldsymbol{w^r}$.

The framework incorporates a nested optimization process to stabilize and accelerate training. During the outer iterations, gradients $\nabla_{\boldsymbol{w}} f(\boldsymbol{x}_{t_i}, \boldsymbol{w^r})$ are computed via backpropagation. These gradients encapsulate the dynamic behavior of the NN during training and provide critical insights into model adjustments. Fixing the gradients enables multiple updates to the parameters on the linearized objective, derived using the current gradient, before recalculating the gradient itself. This periodic fixed-gradient approach significantly reduces the sensitivity of training to noisy financial data and simplifies the complex optimization landscape. The linearized risk measure $\rho(V)$ is approximated as

$$\rho(V) \approx \rho(V^r) + \rho'(V^r) \cdot \nabla_{\boldsymbol{w}} V\big|_{\boldsymbol{w}=\boldsymbol{w^r}} \cdot (\boldsymbol{w} - \boldsymbol{w^r}), \tag{10}$$

where $V^r$ is the PNL value at $\boldsymbol{w^r}$

$$V^r = -Z + \sum_{t_i=0}^{T-1} f(\boldsymbol{x}_{t_i}, \boldsymbol{w^r})\Delta P_{t_i} - c\sum_{t_i=0}^{T-1} \left|f(\boldsymbol{x}_{t_{i+1}}, \boldsymbol{w^r}) - f(\boldsymbol{x}_{t_i}, \boldsymbol{w^r})\right|P_{t_i}. \tag{11}$$

The gradient of the risk measure $\rho(V)$ with respect to $\boldsymbol{w}$ is computed as

$$\nabla_{\boldsymbol{w}}\rho(V) = \rho'(V^r) \cdot \nabla_{\boldsymbol{w}} V\big|_{\boldsymbol{w}=\boldsymbol{w^r}}, \tag{12}$$

where the major complexity comes from the calculation of $\nabla_{\boldsymbol{w}} V\big|_{\boldsymbol{w}=\boldsymbol{w^r}}$ which is also very sensitive to the noise data. Our design is to apply (9) to smooth and simplify the model training dynamic. First, we make smooth approximation on the hedging transaction cost, and define

$$\hat{V} = -Z + \sum_{t_i=0}^{T-1} \hat{f}(\boldsymbol{x}_{t_i}, \boldsymbol{w})\Delta P_{t_i} - c\sum_{t_i=0}^{T-1} \left(\hat{f}(\boldsymbol{x}_{t_{i+1}}, \boldsymbol{w}) - \hat{f}(\boldsymbol{x}_{t_i}, \boldsymbol{w})\right)^2 P_{t_i}, \tag{13}$$

then, we can obtain the gradient approximation as

$$
\begin{aligned}
\nabla_{\boldsymbol{w}}\hat{V} = &\sum_{t_i=0}^{T-1} \nabla_{\boldsymbol{w}} f(\boldsymbol{x}_{t_i}, \boldsymbol{w^r})\Delta P_{t_i} - 2c\sum_{t_i=0}^{T-1} \left[ f(\boldsymbol{x}_{t_{i+1}}, \boldsymbol{w^r})\nabla_{\boldsymbol{w}} f(\boldsymbol{x}_{t_{i+1}}, \boldsymbol{w^r}) - f(\boldsymbol{x}_{t_i}, \boldsymbol{w^r})\nabla_{\boldsymbol{w}} f(\boldsymbol{x}_{t_i}, \boldsymbol{w^r}) \right] P_{t_i} \\
&- 2c\sum_{t_i=0}^{T-1} P_{t_i} \cdot \left[ \nabla_{\boldsymbol{w}} f(\boldsymbol{x}_{t_{i+1}}, \boldsymbol{w^r})\nabla_{\boldsymbol{w}} f(\boldsymbol{x}_{t_{i+1}}, \boldsymbol{w^r})^{\top} - \nabla_{\boldsymbol{w}} f(\boldsymbol{x}_{t_i}, \boldsymbol{w^r})\nabla_{\boldsymbol{w}} f(\boldsymbol{x}_{t_i}, \boldsymbol{w^r})^{\top} \right](\boldsymbol{w} - \boldsymbol{w^r}).
\end{aligned}
\tag{14}
$$

Since both $\nabla_{\boldsymbol{w}} f(\boldsymbol{x}_{t_i}, \boldsymbol{w^r})$ and $\nabla_{\boldsymbol{w}} f(\boldsymbol{x}_{t_{i+1}}, \boldsymbol{w^r})$ are from the backpropagation and fixed for subsequent inner iterations, the calculation of $\nabla_{\boldsymbol{w}}\hat{V}$ becomes linear function w.r.t. model parameter $\boldsymbol{w}$ in the inner iterations.

The nested optimization framework alternates between outer and inner iterations, balancing computational efficiency with training precision, ensuring robust performance even in the presence of noisy data. By holding the gradients constant, the framework linearizes the objective function around the current parameter state $\boldsymbol{w^r}$, significantly reducing the complexity of the optimization landscape. The inner iterations leverage these fixed gradients to solve a simplified optimization problem efficiently. The parameters during these iterations, denoted as $\hat{\boldsymbol{w}}$, are updated iteratively according to

$$
\hat{\boldsymbol{w}}^{j+1} = \hat{\boldsymbol{w}}^j - \eta\rho'(V^r) \cdot \nabla_{\hat{\boldsymbol{w}}^j}\hat{V},
\tag{15}
$$

where $\eta$ represents the learning rate, $\rho'(V^r)$ is the derivative of the risk measure with respect to the PNL value $V^r$, and $\nabla_{\boldsymbol{w}}\hat{V}$ measures the sensitivity of the PNL value to the model parameters in the inner iterations. This periodic fixed-gradient approach ensures that the inner updates are computationally stable and directly aligned with risk minimization objectives.

---

**Algorithm 1** Nested Optimization Procedure with Linearized Objective

---

**Require:** Initial model $\boldsymbol{w^0}$, learning rate $\eta$, convergence tolerance $\epsilon$, outer iterations $R$, inner iterations $N$
**Ensure:** Optimal hedging model $\boldsymbol{w^*}$
 1: **for each outer iteration $r = 0, \ldots, R$ do**
 2:     Compute neural network gradients $\{\boldsymbol{\nabla_w f(x_{t_i}, w^r)}\}_{t_i=0}^{T-1}$ via backpropagation
 3:     Evaluate portfolio value $V^r \leftarrow -Z + \sum_{t_i=0}^{T-1} f(x_{t_i}, w^r)\Delta P_{t_i} - c\sum_{t_i=0}^{T-1} |f(x_{t_{i+1}}, w^r) - f(x_{t_i}, w^r)|P_{t_i}$
 4:     Compute derivative of risk measure $\rho'(V^r)$
 5:     **Initialize: $\hat{\boldsymbol{w}}^0 \leftarrow \boldsymbol{w^r}$**
 6:     **for each inner iteration $j = 0, \ldots, N$ do**
 7:         Compute $\nabla_{\hat{\boldsymbol{w}}^j}\hat{V}$ via (14)
 8:         Update parameters: $\hat{\boldsymbol{w}}^{j+1} = \hat{\boldsymbol{w}}^j - \eta\rho'(V^r) \cdot \nabla_{\hat{\boldsymbol{w}}^j}\hat{V}$
 9:         **if $\|\hat{\boldsymbol{w}}^{j+1} - \hat{\boldsymbol{w}}^j\| < \epsilon$ then**
10:             **break inner loop**
11:         **end if**
12:     **end for**
13:     Update parameters: $\boldsymbol{w}^{r+1} \leftarrow \hat{\boldsymbol{w}}^{j+1}$
14: **end for**

---

The innovative design of the proposed framework offers several significant contributions to dynamic hedging optimization. First, it enhances computational efficiency by decoupling gradient computation from parameter updates. Our design allows for multiple inner iterations to optimize the linearized objective without recalculating gradients via backpropagation, thereby reducing computational overhead. This efficiency is particularly advantageous in high-dimensional and noisy financial environments, where traditional approaches often struggle with scalability. Second, the linearized representation of the objective function mitigates the challenges associated with non-convex loss surfaces. By simplifying the optimization landscape, the framework improves both the robustness and interpretability of the training process. This design ensures that the model achieves stable and reliable performance, even under the volatile and unpredictable conditions characteristic of financial markets. Third, the nested structure introduces a dynamic adaptability that is

critical for financial applications. By maintaining flexibility to accommodate evolving market conditions, the framework demonstrates resilience against non-stationarity and ensures the effectiveness of hedging strategies across a variety of market scenarios.

This nested optimization framework addresses several critical limitations inherent in traditional training approaches for financial applications. The periodic fixed-gradient approach stabilizes the optimization process, reduces sensitivity to noisy data, and accelerates convergence by leveraging a simplified and linearized objective during inner iterations. These features collectively facilitate more robust training dynamics and improved model performance under challenging conditions. For further implementation details, refer to Algorithm 1.

## 4.2 Anchor Hedge Strategy with Black-Scholes Delta Integration

The Anchor Hedge Strategy introduces the Black-Scholes delta as a foundational guide to enhance the robustness and practicality of neural network-based hedging strategies. This approach directly addresses the challenges posed by residual model drift and noisy financial data, providing a flexible yet rigorous mechanism for risk management.

The Black-Scholes delta, denoted as $\delta_{t_i}^{bs}$, measures the sensitivity of an option's price to changes in the underlying asset's price. Derived from the Black-Scholes formula, it accounts for critical factors such as the current price of the underlying asset, the option's strike price, the time remaining until expiration, and the asset's volatility. By incorporating these inputs, the delta serves as a probability-like measure, representing the likelihood of the option ending in the money. This metric is widely recognized as a cornerstone of financial mathematics, offering a robust basis for hedging.

While the proposed optimization framework minimizes risk through a convex objective function, it does not guarantee perfect hedging due to inherent limitations such as market frictions, model drift, and noisy data. To address these residual inefficiencies, the Black-Scholes delta is incorporated as an independent adjustment mechanism, ensuring that the hedging strategy remains aligned with established financial theories and responsive to real-time market dynamics.

This integration is operationalized by defining upper and lower bounds for position adjustments, based on the Black-Scholes delta. Specifically, the boundaries are formulated as

$$
\begin{aligned}
\Delta_{t_i}^l &= \delta_{t_i}^{bs} - f(\boldsymbol{x}_{t_{i+1}}, \boldsymbol{w}), \\
\Delta_{t_i}^u &= \delta_{t_i}^{bs} + f(\boldsymbol{x}_{t_{i+1}}, \boldsymbol{w}).
\end{aligned}
\tag{16}
$$

These outputs are designed to refine the position boundaries by capturing key market dynamics, thus enhancing the overall effectiveness of the strategy. The hedging position at the next time step, $\delta_{t_{i+1}}$, is determined as

$$
\delta_{t_{i+1}} = \min\{\max\{\Delta_{t_i}^l, \delta_{t_i}\}, \Delta_{t_i}^u\}.
\tag{17}
$$

This formulation ensures that the new position remains within the defined transaction band, providing a controlled and adaptive mechanism for adjusting positions. By constraining transactions to these bounds, the strategy mitigates excessive trading, reduces costs, and improves risk management.

## 4.3 Convergence and Robustness Analysis

We examine the convergence behavior and robustness of the proposed nested optimization framework, addressing the challenges posed by dynamic financial markets and noisy data. The analysis focuses on bounding model gradient variations and ensuring the stability of the linearized neural network model outputs during the optimization process.

The nested optimization framework achieves convergence by leveraging a linearized objective function, which simplifies the training dynamics. At the core of this approach lies the transition from the current model parameters $\boldsymbol{w^r}$ to updated parameters $\boldsymbol{w}$. This transition is represented by the auxiliary variable

$$
\boldsymbol{z} = \boldsymbol{w^r} + \tau(\boldsymbol{w} - \boldsymbol{w^r}), \quad 0 \leq \tau \leq 1.
\tag{18}
$$

By analyzing the gradient trajectory along $\boldsymbol{z}$, the optimization process captures the first-order behavior of the NN model

$$\frac{df(\boldsymbol{x}_{t_i}, \boldsymbol{z})}{d\tau} = \nabla_{\boldsymbol{w}} f(\boldsymbol{x}_{t_i}, \boldsymbol{z})^\top (\boldsymbol{w} - \boldsymbol{w^r}), \tag{19}$$

where $f(\boldsymbol{x}_{t_i}, \boldsymbol{z})$ represents the NN model function. Integrating over $\tau \in [0, 1]$, the update dynamics are described as

$$f(\boldsymbol{x}_{t_i}, \boldsymbol{w}) - f(\boldsymbol{x}_{t_i}, \boldsymbol{w^r}) = \int_0^1 \nabla_{\boldsymbol{w}} f(\boldsymbol{x}_{t_i}, \boldsymbol{z})^\top (\boldsymbol{w} - \boldsymbol{w^r}) \, d\tau. \tag{20}$$

The integral can be decomposed to distinguish the contribution of the gradient at $\boldsymbol{w^r}$ and variations along the path

$$f(\boldsymbol{x}_{t_i}, \boldsymbol{w}) - f(\boldsymbol{x}_{t_i}, \boldsymbol{w^r}) = \nabla_{\boldsymbol{w}} f(\boldsymbol{x}_{t_i}, \boldsymbol{w^r})^\top (\boldsymbol{w} - \boldsymbol{w^r}) + \int_0^1 \left( \nabla_{\boldsymbol{w}} f(\boldsymbol{x}_{t_i}, \boldsymbol{z}) - \nabla_{\boldsymbol{w}} f(\boldsymbol{x}_{t_i}, \boldsymbol{w^r}) \right)^\top (\boldsymbol{w} - \boldsymbol{w^r}) \, d\tau. \tag{21}$$

To bound the higher-order term, the Cauchy-Schwarz inequality is applied

$$\left| \int_0^1 \left( \nabla_{\boldsymbol{w}} f(\boldsymbol{x}_{t_i}, \boldsymbol{z}) - \nabla_{\boldsymbol{w}} f(\boldsymbol{x}_{t_i}, \boldsymbol{w^r}) \right)^\top (\boldsymbol{w} - \boldsymbol{w^r}) \, d\tau \right| \leq \int_0^1 \left\| \nabla_{\boldsymbol{w}} f(\boldsymbol{x}_{t_i}, \boldsymbol{z}) - \nabla_{\boldsymbol{w}} f(\boldsymbol{x}_{t_i}, \boldsymbol{w^r}) \right\|_2 \cdot \left\| \boldsymbol{w} - \boldsymbol{w^r} \right\|_2 \, d\tau. \tag{22}$$

The robustness of the nested optimization framework stems from its ability to fix gradients during inner iterations, significantly reducing sensitivity to noisy data and ensuring stability throughout the training process. Anchoring the training dynamics to these fixed gradients allows the linearized model outputs, $\hat{f}$, to effectively filter out high-dimensional noise, thereby facilitating meaningful and precise parameter updates.

The Hessian matrix $\nabla^2 f(\boldsymbol{x}_{t_i}, \boldsymbol{w^r})$ quantifies the second-order behavior of the model function by measuring the rate of change of the gradient with respect to the parameters. A bounded Hessian ensures smooth changes in the gradient, a crucial property for maintaining the stability and accuracy of linear approximations. This smoothness prevents abrupt shifts in the optimization trajectory, which can destabilize training in the presence of noise.

To analyze gradient dynamics, consider the difference between the gradients at the current parameter state $\boldsymbol{w^r}$ and a perturbed parameter $\boldsymbol{z}$

$$\left\| \nabla f(\boldsymbol{x}_{t_i}, \boldsymbol{w^r}) - \nabla f(\boldsymbol{x}_{t_i}, \boldsymbol{z}) \right\| \approx \left\| \boldsymbol{z} - \boldsymbol{w^r} \right\| \cdot \left\| \nabla^2 f(\boldsymbol{x}_{t_i}, \boldsymbol{w^r}) \right\|. \tag{23}$$

This expression demonstrates that the gradient change is proportional to the perturbation $\|\boldsymbol{z} - \boldsymbol{w^r}\|$ and the magnitude of the Hessian. A bounded Hessian ensures that the gradient change remains controlled, even under noisy data conditions, preserving the framework's stability.

For small perturbations $\boldsymbol{w} - \boldsymbol{w^r}$, the relative change in the gradient is similarly bounded

$$\frac{\left\| \nabla f(\boldsymbol{x}_{t_i}, \boldsymbol{w^r}) - \nabla f(\boldsymbol{x}_{t_i}, \boldsymbol{z}) \right\|}{\left\| \nabla f(\boldsymbol{x}_{t_i}, \boldsymbol{w^r}) \right\|} \leq \frac{\left\| \boldsymbol{z} - \boldsymbol{w^r} \right\| \cdot \left\| \nabla^2 f(\boldsymbol{x}_{t_i}, \boldsymbol{w^r}) \right\|}{\left\| \nabla f(\boldsymbol{x}_{t_i}, \boldsymbol{w^r}) \right\|}. \tag{24}$$

This bounded ratio quantifies the stability of the gradient dynamics, ensuring that even under noisy data, the optimization remains resilient.

To further illustrate this robustness, we introduce the auxiliary variable $\boldsymbol{z} = \boldsymbol{w^r} + \tau(\boldsymbol{w} - \boldsymbol{w^r})$, where $\tau \in [0, 1]$. Along this trajectory, the gradient difference is bounded as

$$\left\| \nabla f(\boldsymbol{x}_{t_i}, \boldsymbol{z}) - \nabla f(\boldsymbol{x}_{t_i}, \boldsymbol{w^r}) \right\| \leq \tau \beta \left\| \boldsymbol{w} - \boldsymbol{w^r} \right\|, \tag{25}$$

where $\beta$ denotes the upper bound on the Hessian, with a detailed derivation provided in Appendix D.

By anchoring updates to a fixed gradient during inner iterations and ensuring that the Hessian remains bounded, the framework achieves a high level of resilience against noisy data. This robustness enables stable optimization, isolates the effects of high-dimensional noise, and preserves the integrity of parameter updates. These properties make the framework particularly effective in dynamic financial environments characterized by noise and volatility.

## 5 Numerical Results

To evaluate the effectiveness of our proposed DHLNN, we conducted comprehensive numerical simulations and experiments, comparing its performance against three alternative strategies, i.e., Deep Hedging with Multi-Layer Perceptron (DHMLP), Deep Hedging with Neural Tangent Bootstrap (DHNTB), and the Black-Scholes Delta Hedging (BSDH) strategy. The BSDH approach, a foundational method in options trading, involves continuously adjusting the holdings in the underlying asset to maintain a delta-neutral portfolio (Hull & Basu, 2016; Broadie et al., 1999), thereby minimizing exposure to price fluctuations. It serves as a standard benchmark in our experiments. In addition to BSDH, we incorporated two state-of-the-art deep hedging baselines, DHMLP and DHNTB, which represent advanced solutions for dynamic hedging (Buehler et al., 2019; Imaki et al., 2021). These baselines underscore the progression of deep learning techniques in financial risk management and provide a critical context for evaluating our method.

### 5.1 Experimental Settings

The DHLNN architecture is built with four hidden layers, each comprising 64 neurons. To capture complex nonlinear relationships in financial data, each neuron employs the ReLU activation function, which introduces nonlinearity while maintaining computational simplicity. The model training process is conducted with a learning rate of 0.001 to ensure stable and efficient convergence. The Adam optimizer is utilized for backpropagation during the outer iterations, leveraging its adaptive learning capabilities for the adjusted gradient calculation, where the exponential decay rates for the first and second moment estimates are set to $\beta_1 = 0.9$ and $\beta_2 = 0.999$, respectively, while a small constant $\epsilon = 10^{-8}$ is added to enhance numerical stability. The adjusted gradient by Adam in the backpropagation are then fixed during the nested linearized model updating process with the same learning rate, ensuring consistency and robustness in the optimization.

To comprehensively evaluate the performance of our proposed deep hedging strategy, we utilized two types of datasets, i.e., simulated price trajectories and real market data. Additionally, two distinct types of options, European Call Options and Lookback Call Options, were employed to assess the robustness of our approach under varying market conditions.

The simulated price trajectories in our experiments are generated using a geometric Brownian motion framework, a standard approach for modeling stochastic price dynamics in financial markets. The price evolution is defined as

$$dP_{t_i} = \mu P_{t_i} dt + \sigma P_{t_i} dW_{t_i}, \tag{26}$$

where $P_{t_i}$ denotes the price of the underlying asset at time $t_i$, $\mu$ represents the drift term, $\sigma$ is the volatility, and $dW_{t_i}$ corresponds to an increment of a Wiener process. The drift term is set to $\mu = 0$, ensuring the price follows a martingale process, which is a common assumption in derivative pricing and hedging frameworks (Shreve, 2005). The simulation spans a maturity period of $T = 30/365$ years, i.e., 30 days, with daily time steps defined as $d_t = 1/365$. At each time step, stochastic increments $dW_{t_i}$ are modeled as standard normal random variables scaled by $\sqrt{d_t}$. A correction term, $\frac{\sigma^2 t_i}{2}$, is incorporated to adjust for the expected drift, maintaining consistency with risk-neutral pricing dynamics. The resulting price evolution is expressed as

$$P_{t_i} = P_0 \exp\left(B_{t_i} - \frac{\sigma^2 t_i}{2}\right), \tag{27}$$

where $B_{t_i}$ represents the cumulative Brownian motion.

In the experiments, the volatility parameter $\sigma$ is fixed at 0.1 during training to ensure a stable and controlled environment for optimizing the hedging model (Gatheral, 2011). For testing, higher volatility scenarios

$\sigma \in \{0.2, 0.3, 0.4\}$ are introduced to evaluate the model's robustness and adaptability to out-of-distribution conditions. The initial price $P_0$ is normalized to 1, ensuring consistency across simulations. A total of $10,000$ independent price trajectories are generated in each epoch for training and testing to provide diverse and representative datasets for evaluation.

The choice of experimental parameters reflects both practical relevance and alignment with established practices in financial research. The transaction costs, ranging from $2 \times 10^{-3}$ to $8 \times 10^{-3}$, capture realistic market frictions observed in trading environments, allowing the evaluation of hedging strategies under varying cost pressures (Gârleanu & Pedersen, 2013). Similarly, the strike prices of 1.0 and 1.2 enable the analysis of hedging performance across different market conditions (Björk, 2009). A strike price of 1.0 corresponds to an at-the-money scenario, where the derivative's value is highly sensitive to market movements, while 1.2 represents an out-of-the-money condition, characterized by lower immediate risk but potentially higher exposure over time. These choices ensure the evaluation covers a broad spectrum of risk exposures and market conditions.

The volatility levels, spanning 0.1 to 0.4, are chosen to reflect diverse market dynamics, from relatively stable to highly volatile environments. Volatility is a critical factor influencing derivative pricing and hedging, and assessing performance under varying volatility regimes provides a comprehensive understanding of the model's robustness. The inclusion of out-of-distribution testing scenarios further underscores the generalizability of the proposed framework, as it evaluates performance in conditions beyond the training data distribution.

For real market data, we used detailed financial records encompassing various stocks across multiple sectors (Andrew Meyer, 2021). This dataset includes an order book, which captures real-time buy and sell orders organized by price levels, providing insights into market demand and supply. From this data, we derived critical metrics such as the bid-ask spread, an indicator of trading costs, and the WAP, a comprehensive valuation metric accounting for both price and volume. WAP served as the basis for defining price trajectories and assessing the statistical volatility of assets, offering a real-world benchmark for evaluating our hedging strategy.

We tested our strategy using two types of options, i.e., the European Call Option and the Lookback Call Option. These derivatives were chosen to evaluate the strategy under scenarios of increasing complexity. The European Call Option is a standard derivative that provides the holder the right, but not the obligation, to purchase the underlying asset at a predetermined strike price at maturity. This straightforward derivative allows us to evaluate the strategy's ability to manage risk in a scenario where the primary concern is the asset's price at maturity. The Lookback Call Option introduces a higher level of complexity. Its payoff depends on the maximum price of the underlying asset during its life, making it a path-dependent derivative. This option is particularly sensitive to the entire price trajectory rather than just the price at maturity, providing a stringent test of the strategy's robustness. By incorporating this option, we evaluated the method's effectiveness in scenarios involving significant price volatility and path dependency, offering deeper insights into its real-world applicability.

### 5.2 Metrics for Performance Evaluation

PNL serves as a foundational metric for assessing the performance of hedging strategies. It quantifies the financial outcome of trading activities relative to liabilities incurred from derivative payoffs. A key objective of any hedging framework is to achieve a PNL value close to zero, indicating effective risk mitigation and minimal residual exposure.

Deviations in PNL from neutrality reveal potential inefficiencies. A PNL significantly above zero may indicate over-hedging, which could imply unintended speculation or excessive trading. Conversely, a negative PNL suggests under-hedging, where the portfolio fails to sufficiently cover the derivative's payoff, exposing it to significant financial losses. Both scenarios highlight the importance of carefully balancing hedging effectiveness, cost efficiency, and risk control. A desirable PNL distribution clusters around zero with minimal variance, demonstrating the strategy's consistency and neutrality in managing financial risks.

To evaluate the effectiveness of the hedging strategy more comprehensively, we employ two additional metrics, i.e., Expected Shortfall and Entropic Loss. These metrics focus on the downside risk by emphasizing negative

PNL values and provide deeper insights into the strategy's risk profile and robustness. Smaller values for both metrics correspond to better performance in managing risks.

In the context of hedging, perfect hedging is the ideal scenario where the portfolio's value exactly offsets the derivative's payoff at maturity, yielding a PNL of zero. However, due to market frictions, transaction costs, and model imperfections, achieving perfect hedging is practically unattainable. Instead, the goal is to minimize residual risks, as reflected in the distribution of PNL values. A well-performing strategy ensures that the PNL distribution is centered near zero with minimal variance, reducing both over-hedging and under-hedging risks.

Expected Shortfall is a widely used tail risk measure that evaluates the average loss within the worst $(1 - \alpha)\%$ of outcomes defined as

$$\text{Expected Shortfall}_\alpha = \mathbb{E}[-\text{PNL} \mid -\text{PNL} \geq \text{VaR}_\alpha], \tag{28}$$

where $\text{VaR}_\alpha$, i.e., Value-at-Risk, represents the loss threshold at the confidence level $\alpha$, satisfying $P(-\text{PNL} \geq \text{VaR}_\alpha) = 1 - \alpha$. Expected Shortfall provides a more comprehensive view of tail risk compared to VaR by averaging losses beyond the threshold, capturing the severity of extreme losses. A lower Expected Shortfall indicates effective tail risk management, aligning the PNL distribution closer to zero and mitigating the impact of adverse market conditions. We choose $\alpha = 0.9$ for Expected Shortfall measurement in our experiments.

Entropic Loss offers a utility-based perspective on risk by assigning greater penalties to larger losses through exponential weighting. It is defined as

$$\text{Entropic Loss} = \frac{1}{\hat{\alpha}} \log \mathbb{E}\left[e^{-\hat{\alpha} \cdot (-\text{PNL})}\right], \tag{29}$$

where $\hat{\alpha} > 0$ is the risk aversion parameter, and we set $\hat{\alpha} = 1.0$. The exponential term $e^{-\hat{\alpha} \cdot (-\text{PNL})}$ amplifies the impact of larger losses, making Entropic Loss particularly sensitive to extreme outcomes. A smaller Entropic Loss reflects improved robustness against significant financial setbacks, demonstrating the strategy's ability to withstand severe market conditions.

## 5.3 Experiments with Simulated Market Data

### 5.3.1 Convergence Performance with Simulated Market Data

Fig. 2 compares the convergence performance of different hedging strategies, including the proposed DHLNN, with baseline methods DHNTB, DHMLP, and BSDH. The experiments were conducted on simulated market data for a European call option with a strike price of 1.2, a fixed volatility of 0.1, and a transaction cost rate of $2 \times 10^{-3}$. The performance metric is the PNL distribution, where a narrower and more centered distribution around zero signifies better hedging performance by minimizing liability discrepancies and achieving a near risk-neutral position.

The results demonstrate that DHLNN consistently outperforms baseline methods across different training epochs. Its PNL distributions become progressively sharper and more concentrated around zero as training advances, reflecting superior convergence and risk mitigation capabilities. At just 10 training epochs, DHLNN already exhibits a significantly narrower and more centered PNL distribution than the baselines as shown in Fig. 2(a). With increasing training epochs, the performance gap widens further, and by epoch 40 as shown in Fig. 2(d), DHLNN achieves near-perfect convergence, underscoring its computational efficiency and robust learning dynamics.

In contrast, the baseline methods show slower improvements and less stable convergence. BSDH, while maintaining a relatively stable PNL distribution due to its reliance on fixed hedging rules, fails to adapt to dynamic market conditions. DHNTB and DHMLP, though more flexible, exhibit broader PNL distributions with higher variances, indicating greater exposure to over-hedging and under-hedging risks. These results highlight the limitations of traditional and baseline deep learning approaches while showcasing the effectiveness of DHLNN in addressing key challenges in dynamic hedging.

Fig. 3 illustrates a detailed comparison of the performance of deep hedging models, including DHLNN, DHNTB, DHMLP, and BSDH, across training epochs as10, 20, 30, and 40. The purpose of this experiment is

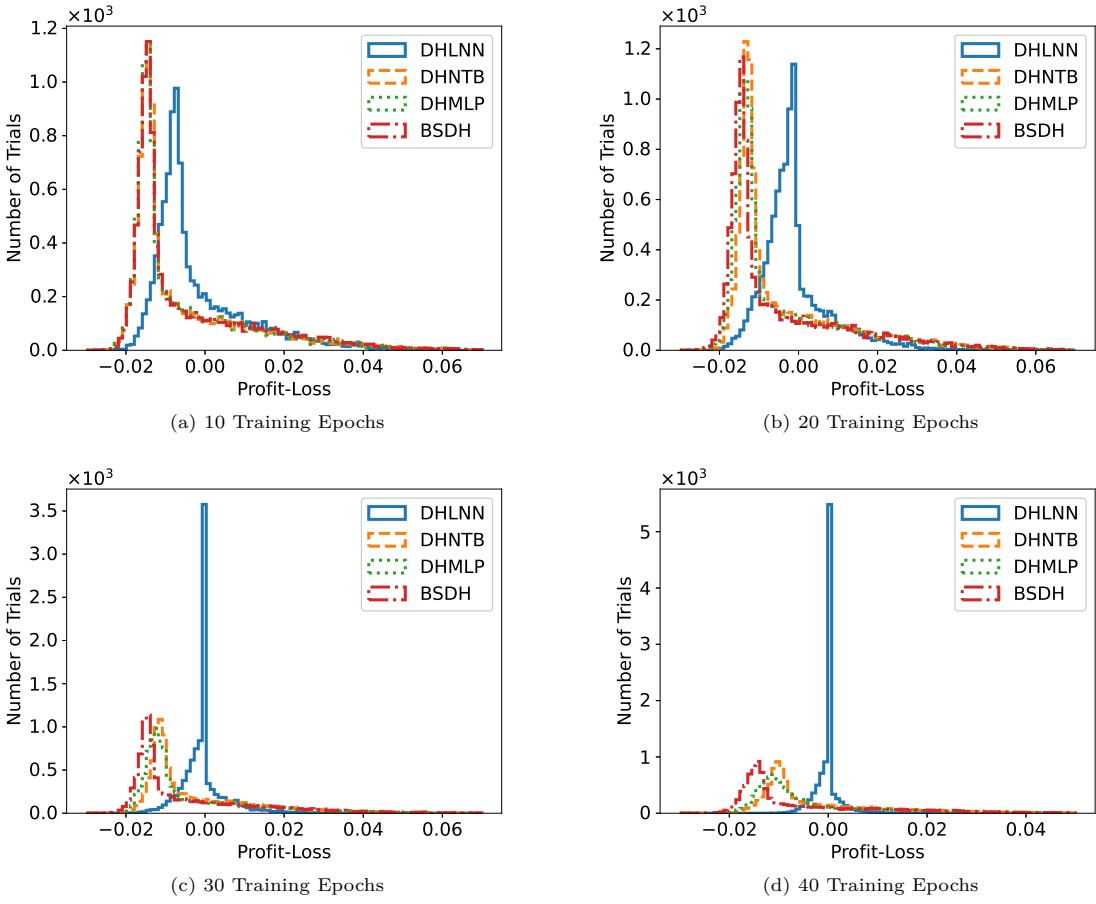

Figure 2: Comparison of convergence performance for deep hedging models across varying training epochs $\{10, 20, 30, 40\}$ on a European option with a strike price of 1.2. The experiment is conducted under a fixed volatility of 0.1 and transaction costs of $2 \times 10^{-3}$. The optimal PNL distribution should be narrow and centered around zero, indicating minimal liability discrepancy, robust convergence, and efficient hedging performance with low sensitivity to noise and transaction costs.

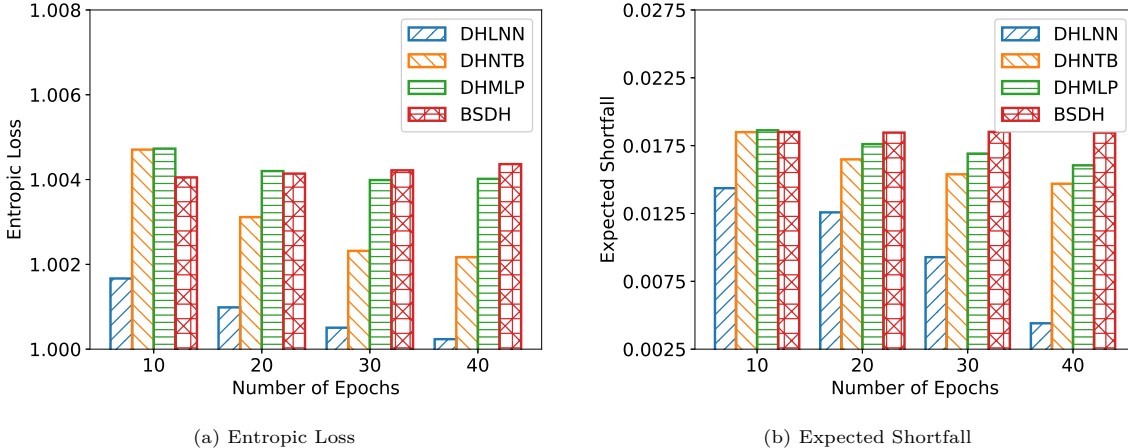

(a) Entropic Loss  (b) Expected Shortfall

Figure 3: Performance comparison of deep hedging models based on Entropic Loss (a) and Expected Shortfall (b) metrics across training epochs 10, 20, 30, and 40. The experiments are conducted on a European call option with a strike price of 1.2, a transaction cost rate of $2 \times 10^{-3}$, and a fixed volatility of 0.1. Entropic Loss quantifies overall risk, with lower values indicating better uncertainty management. Expected Shortfall measures tail risk, with smaller values reflecting reduced exposure to extreme losses. DHLNN consistently outperforms baseline methods DHNTB, DHMLP, and BSDH across both metrics, demonstrating faster convergence, superior robustness, and more effective hedging under dynamic market conditions.

to evaluate the proposed DHLNN framework against baseline methods using Entropic Loss and Expected Shortfall. These metrics capture the model's effectiveness in managing overall and tail risks, respectively, and provide a comprehensive assessment of each model's ability to handle the complexities of financial risk management. By analyzing performance over multiple training epochs, the experiment underscores the capability of DHLNN to achieve superior risk minimization, faster convergence, and enhanced robustness, offering a more efficient and practical solution for financial markets.

The Entropic Loss, as shown in Fig. 3(a), emphasizes the importance of uncertainty and extreme losses in evaluating portfolio performance. Across all epochs, DHLNN consistently achieves the lowest Entropic Loss, reflecting its superior ability to mitigate amplified risks and manage uncertainties. As training progresses, the significant reduction in Entropic Loss highlights DHLNN's efficient convergence and ability to stabilize training. In contrast, the baseline methods, including DHNTB and DHMLP, show slower improvement and remain less effective in minimizing risk. BSDH, as a static rule-based approach, demonstrates negligible improvement, further highlighting its limitations in dynamic financial environments.

Expected Shortfall, depicted in Fig. 3(b), provides complementary insights by measuring the average loss in the worst-case scenarios, capturing the ability of models to manage tail risks. Similar to Entropic Loss, DHLNN consistently achieves the lowest values across all epochs, demonstrating its effectiveness in minimizing extreme losses. The steady decrease in Expected Shortfall for DHLNN over training epochs indicates its ability to converge quickly to optimal hedging strategies. By comparison, the baseline methods exhibit slower convergence rates and higher exposure to tail risks, highlighting their reduced efficacy under adverse market conditions.

### 5.3.2 Robustness to Market Frictions with Simulated Market Data

Fig. 4 illustrates the performance of various deep hedging models, including DHLNN, DHNTB, DHMLP, and BSDH, across two transaction cost scenarios, i.e., $2 \times 10^{-3}$ and $4 \times 10^{-3}$. The analysis, conducted for a European option with a strike price of 1.0 after 50 training epochs, focuses on the PNL distributions to assess the models' robustness in handling market frictions.

Under the lower transaction cost as $2 \times 10^{-3}$ shown in Fig. 4(a), the DHLNN model achieves a highly concentrated PNL distribution centered near zero. This result indicates its superior capability to maintain near-risk-neutral hedging while effectively accounting for transaction costs. In contrast, the baseline methods, including DHNTB, DHMLP, and BSDH, display broader PNL distributions with higher variance. These

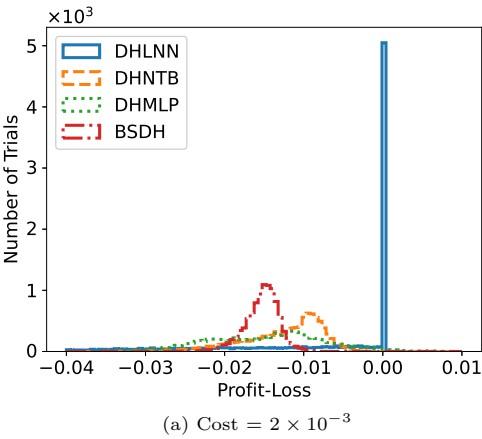 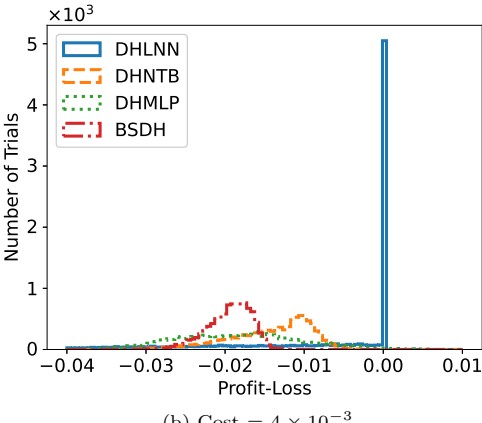

(a) Cost $= 2 \times 10^{-3}$          (b) Cost $= 4 \times 10^{-3}$

Figure 4: Evaluation of hedging performance across different transaction costs ($2 \times 10^{-3}$ and $4 \times 10^{-3}$) for a European option with a strike price of 1.0. The analysis is conducted over 50 training epochs, focusing on the distribution of hedging PNL under varying transaction cost rates, with a fixed volatility of 0.1. A narrower and more concentrated PNL distribution around zero indicates better risk-neutrality and robustness against market frictions.

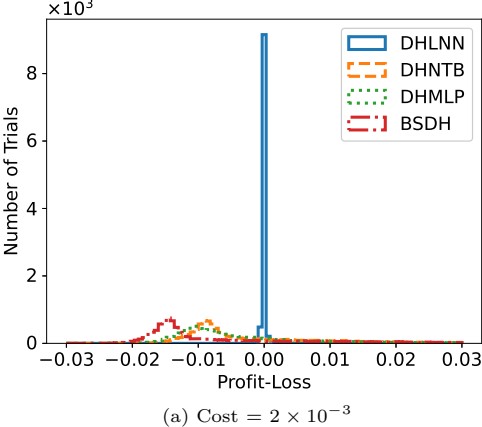 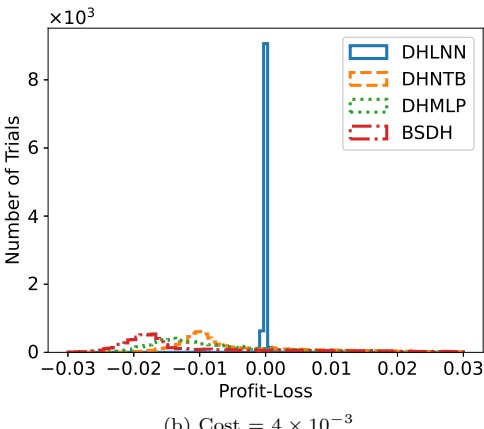

(a) Cost $= 2 \times 10^{-3}$          (b) Cost $= 4 \times 10^{-3}$

Figure 5: Comparison of hedging performance across different transaction costs $2 \times 10^{-3}, 4 \times 10^{-3}$ for a European option with a strike price of 1.2. The analysis, conducted over 50 training epochs, evaluates the distribution of hedging PNL to assess the robustness of different deep hedging models, with volatility set at 0.1. The narrower and more centered PNL distribution reflects better hedging efficiency and risk neutrality.

results reflect the baseline methods' greater exposure to over-hedging and under-hedging risks, with BSDH demonstrating the least adaptability due to its reliance on static hedging rules.

As transaction costs increase to $4 \times 10^{-3}$, shown in Fig. 4(b), DHLNN continues to outperform the baselines by maintaining a stable and narrow PNL distribution around zero. This demonstrates the model's robustness in managing higher market frictions without significant performance degradation. However, the baseline methods exhibit noticeable performance declines under these conditions. Both DHNTB and DHMLP show broader distributions with increased variance, reflecting their struggles in mitigating the impact of higher transaction costs. BSDH, which lacks mechanisms to account for transaction costs, performs the worst, with its PNL distribution further diverging from zero.

Fig. 5 examines the hedging performance of DHLNN and three baseline methods, DHNTB, DHMLP, and BSDH, under transaction costs of $2 \times 10^{-3}$ and $4 \times 10^{-3}$ for a European option with a strike price of 1.2. The experiments, conducted over 50 training epochs, focus on analyzing the PNL distributions to evaluate the robustness and efficiency of each model in managing risk while accounting for market frictions.

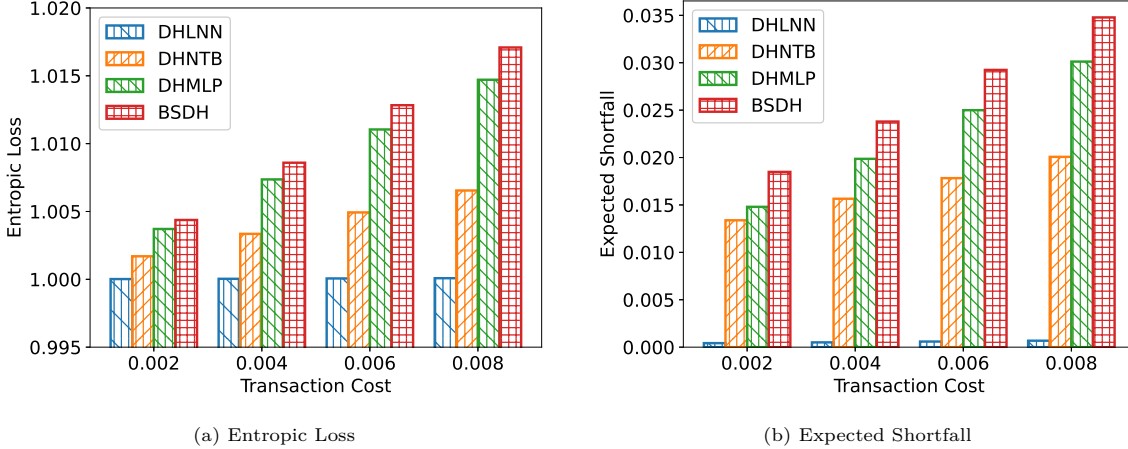

(a) Entropic Loss                    (b) Expected Shortfall

Figure 6: Hedging performance comparison across transaction costs $2 \times 10^{-3}, 4 \times 10^{-3}, 6 \times 10^{-3}, 8 \times 10^{-3}$ for a European option with a strike price of 1.2. The analysis evaluates the Entropic Loss and Expected Shortfall of the hedging PNL after 50 training epochs, where the underlying asset volatility is fixed at 0.1.

With the lower transaction cost $2 \times 10^{-3}$ as shown in Fig. 5(a), DHLNN demonstrates a highly concentrated PNL distribution around zero, indicating its superior capability to neutralize risk while maintaining minimal liability discrepancies. The narrow distribution reflects the model's ability to achieve effective risk mitigation, even with moderate transaction costs. In contrast, the baseline models, DHNTB and DHMLP, display broader distributions with greater variance, highlighting their reduced ability to handle transaction costs effectively. BSDH exhibits the weakest performance, with the broadest PNL distribution and a noticeable divergence from zero, underscoring its limitations as a static, rule-based approach.

Under the higher transaction cost scenario as $4 \times 10^{-3}$ depicted in Fig. 5(b), DHLNN continues to exhibit robust performance, maintaining a concentrated and stable PNL distribution near zero. This stability demonstrates the model's resilience to increased market frictions, showcasing its adaptability and efficiency in challenging environments. Conversely, the performance of DHNTB and DHMLP degrades significantly under higher transaction costs, as reflected in their broader distributions and increased exposure to over-hedging and under-hedging risks. BSDH remains the least effective, with minimal improvement in performance and the widest PNL distribution.

Fig. 6 provides a detailed comparison of the performance of DHLNN and baseline methods over varying transaction costs using Entropic Loss and Expected Shortfall. In Fig. 6(a), the Entropic Loss trends illustrate the models' performance under transaction costs ranging from $2 \times 10^{-3}$ to $8 \times 10^{-3}$. DHLNN consistently achieves the lowest Entropic Loss across all transaction cost levels, demonstrating its superior ability to stabilize training dynamics and minimize the uncertainty in hedging PNL. Even as transaction costs increase, the incremental rise in Entropic Loss for DHLNN remains minimal compared to the other methods, showcasing its robustness against market frictions. In contrast, DHNTB and DHMLP exhibit significantly higher Entropic Loss as transaction costs rise, reflecting their vulnerability to increased trading costs. BSDH consistently underperforms, with the highest Entropic Loss across all transaction cost levels, underscoring its limitations as a static, rule-based approach.

Fig. 6(b) presents the Expected Shortfall trends, offering insights into the models' ability to handle tail risk. Once again, DHLNN outperforms its counterparts by maintaining the lowest Expected Shortfall across all transaction cost scenarios. This demonstrates its ability to manage extreme losses effectively, even under adverse market conditions. As transaction costs increase, the Expected Shortfall for DHNTB and DHMLP grows more steeply compared to DHLNN, indicating less efficient risk mitigation strategies. BSDH remains the least effective, with the highest Expected Shortfall values, further emphasizing its inability to adapt to dynamic market conditions and rising transaction costs.

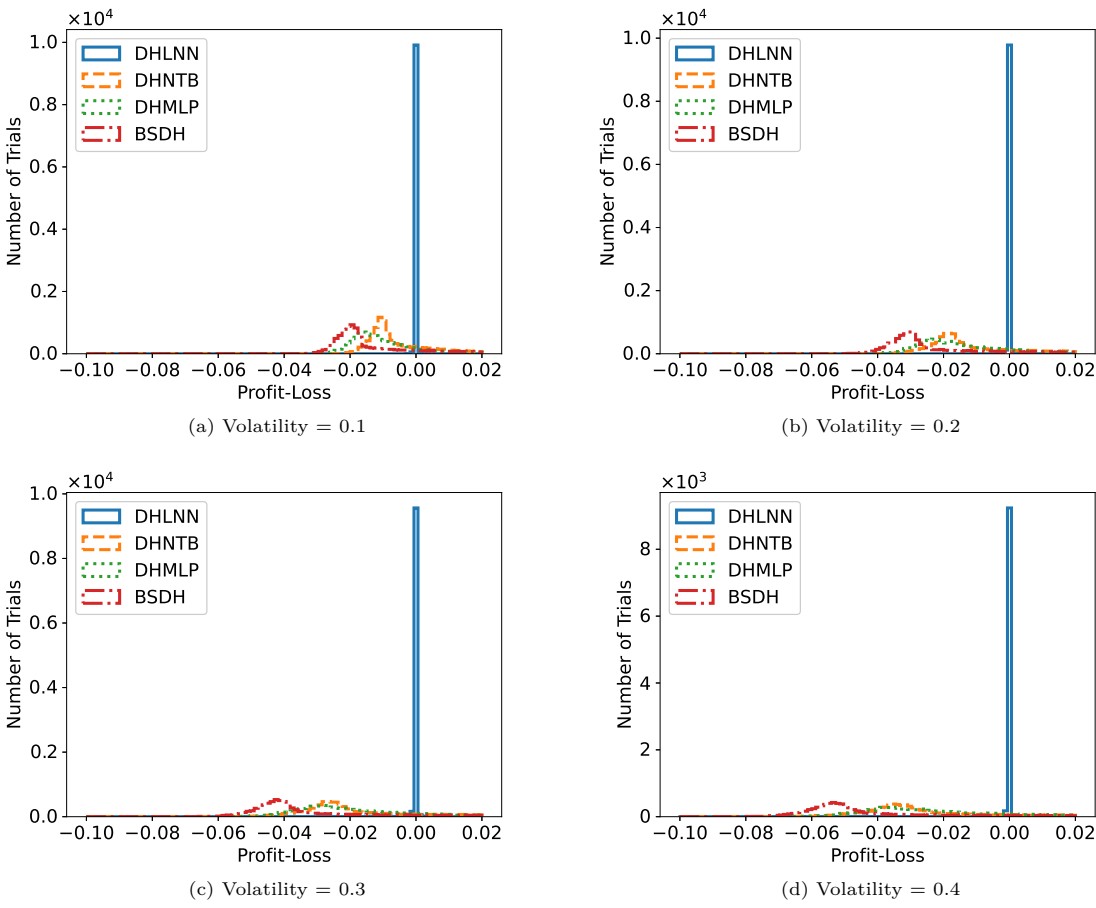

Figure 7: Comparison of hedging performance based on PNL distributions across varying volatility levels of the underlying asset price {0.1, 0.2, 0.3, 0.4} for a European option with a strike price of 1.2. The training data is generated with a fixed volatility of 0.1, while the test data reflects the higher volatility levels. The results highlight DHLNN's resilience, as it maintains concentrated and stable PNL distributions, outperforming DHNTB, DHMLP, and BSDH under more volatile market conditions.

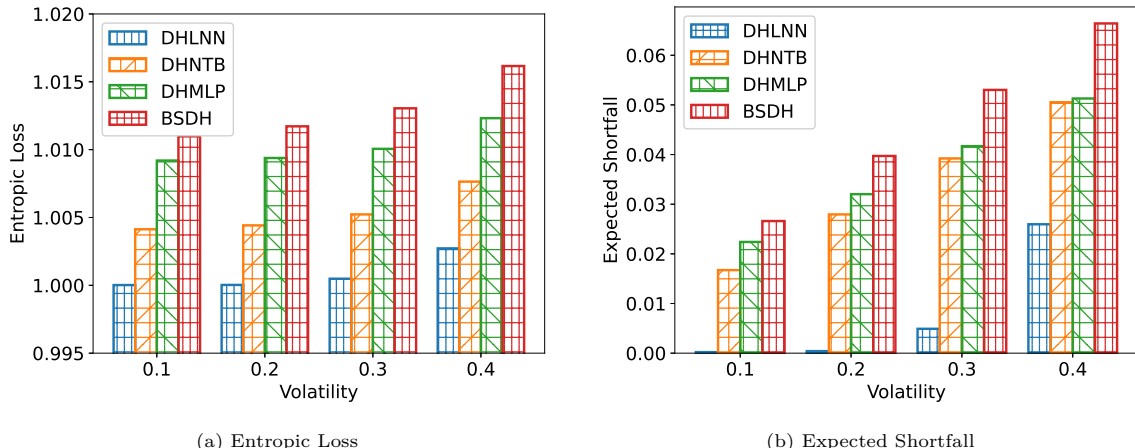

(a) Entropic Loss                    (b) Expected Shortfall

Figure 8: Comparison of hedging performance across varying volatilities of the underlying asset $\{0.1, 0.2, 0.3, 0.4\}$ for a European option with a strike price of 1.2. The analysis evaluates Entropic Loss and Expected Shortfall of the hedging PNL. The results are based on simulations with a transaction cost of $5 \times 10^{-3}$ and 50 training epochs. This experiment highlights the robustness of DHLNN in managing overall and tail risks under dynamic and volatile conditions.

### 5.3.3 Robustness Analysis with Increased Test Volatility in Simulated Market Data

The robustness of hedging models in dynamic and volatile market environments is a key consideration for effective risk management. This experiment evaluates the resilience and adaptability of various hedging strategies, including DHLNN, DHNTB, DHMLP, and BSDH, under increasing levels of test volatility. The training data is generated with a fixed volatility of 0.1 to simulate stable market conditions, while the test data reflects elevated volatility levels of 0.1, 0.2, 0.3, and 0.4. The purpose of this experiment is to assess each model's ability to maintain stable and concentrated PNL distributions under heightened market uncertainties, a critical requirement for robust financial performance. By subjecting the models to test conditions that deviate from their training environments, the experiment provides valuable insights into their generalizability and robustness.

As shown in Fig. 7(a), DHLNN demonstrates superior hedging performance, achieving a highly concentrated PNL distribution centered near zero at the baseline volatility level of 0.1. This indicates its ability to achieve near risk-neutral positions while effectively minimizing liability discrepancies. In contrast, DHNTB and DHMLP show broader PNL distributions, reflecting greater variability in their hedging strategies and increased susceptibility to both over-hedging and under-hedging. BSDH, constrained by its static rule-based approach, performs noticeably worse, with a dispersed PNL distribution indicative of its limited adaptability even in relatively stable conditions.

As the test volatility increases to 0.2 shown in Fig. 7(b), DHLNN maintains its sharp and narrow PNL distribution, highlighting its robustness and ability to adapt to moderate market fluctuations. In comparison, DHNTB and DHMLP exhibit significant broadening of their PNL distributions, signaling reduced robustness and an inability to effectively manage increased uncertainty. BSDH further deteriorates, with its PNL distribution spreading considerably, underscoring its inherent limitations in dynamic environments.

At higher volatility levels of 0.3 and 0.4, the advantages of DHLNN become even more pronounced. Despite the increasingly volatile test conditions, DHLNN continues to deliver concentrated PNL distributions near zero, demonstrating its superior resilience and adaptability. This consistent performance underscores the effectiveness of DHLNN's innovative design, including its linearized training dynamics and periodic fixed-gradient optimization, in managing extreme market fluctuations. In stark contrast, DHNTB and DHMLP struggle to maintain stability, with their PNL distributions becoming increasingly dispersed and less centered. BSDH exhibits the poorest performance, with its PNL distribution further widening, reflecting its inability to adapt to heightened market volatility.

As shown in Fig. 8 this experiment aims to assess the robustness and adaptability of various hedging models, including DHLNN, DHNTB, DHMLP, and BSDH, under increasing levels of market volatility. By examining

the Entropic Loss and Expected Shortfall metrics, the analysis provides insights into the models' capabilities in managing overall and tail risks, respectively. The training is conducted under relatively stable market conditions with volatility as 0.1, while the test data simulates more volatile environments with volatilities ranging from 0.1 to 0.4.

As depicted in Fig. 8(a), DHLNN consistently achieves the lowest Entropic Loss across all volatility levels, reflecting its superior ability to mitigate overall risk and maintain stability in hedging PNL. Even as the test volatility increases, the Entropic Loss for DHLNN remains significantly lower than that of the baseline methods, highlighting its robustness to dynamic market conditions. In contrast, DHNTB and DHMLP exhibit a notable increase in Entropic Loss with higher volatility levels, indicating their reduced effectiveness in managing risk under such conditions. BSDH performs the worst, with Entropic Loss increasing substantially as volatility rises, underscoring its limitations as a static, rule-based approach.

Fig. 8(b) illustrates the Expected Shortfall, which captures the models' abilities to manage tail risks under extreme market conditions. DHLNN again demonstrates its robustness, consistently achieving the lowest Expected Shortfall values across all volatility levels. This indicates its capability to minimize extreme losses and maintain reliable performance even in highly volatile environments. DHNTB and DHMLP, while showing some ability to adapt, exhibit significantly higher Expected Shortfall values compared to DHLNN, particularly at higher volatility levels. BSDH's performance further deteriorates, with its Expected Shortfall values rising steeply, reflecting its inability to handle tail risks effectively.

The results conclusively demonstrate that DHLNN outperforms the baseline methods in managing both overall and tail risks under increasing market volatility. Its innovative design, incorporating linearized neural network training and periodic fixed-gradient optimization, enables it to maintain robust and reliable performance even in challenging market conditions. These findings emphasize the practicality and scalability of DHLNN as a robust solution for financial risk management in dynamic and volatile environments.

## 5.4 Experiments with Real Market Data

Real market data provides a robust testing ground for examining the adaptability and efficiency of various hedging methods under genuine market conditions. These experiments enable us to evaluate the performance of deep hedging frameworks, including DHMLP, DHNTB, and our proposed DHLNN, in managing the intricate dynamics of real financial markets.

A particularly challenging instrument to hedge in real market data is the Lookback option, whose payoff is based on the maximum price of the underlying asset over its lifespan. This path dependency requires sophisticated hedging strategies that consider not just the current price but also the historical price trajectory of the underlying asset. Traditional approaches, such as Black-Scholes Delta Hedging (BSDH), are limited in this context as they rely solely on continuous rebalancing based on the current asset price, neglecting the cumulative price history that determines the payoff. Our analysis highlights the effectiveness of DHLNN compared to DHMLP and DHNTB, emphasizing its ability to adapt to real-world financial complexities and deliver robust hedging performance in managing Lookback options.

### 5.4.1 Convergence Analysis for Lookback Options with Real Market Data

The results in Fig. 9 illustrate the convergence performance of deep hedging models, including DHLNN, DHNTB, and DHMLP, across varying training epochs $\{10, 20, 30, 40\}$ when applied to a Lookback option. The goal of this experiment is to evaluate the ability of each model to adapt to this complexity and achieve a PNL distribution that is narrow and centered around zero. Such a distribution signifies effective risk management by minimizing liability discrepancies and achieving a near risk-neutral hedging position.

At 10 training epochs as shown in Fig. 9(a), DHLNN already demonstrates a concentrated PNL distribution close to zero, outperforming the baseline methods. DHNTB and DHMLP exhibit broader distributions with higher variance, indicating less effective risk mitigation. This highlights the efficiency of DHLNN in stabilizing the hedging process even with limited training. As the training progresses to 20 and 30 epochs as shown in Fig. 9(b) and Fig. 9(c), DHLNN shows further refinement in its PNL distribution, with the peak around zero becoming sharper and variance decreasing significantly. This demonstrates the model's ability to

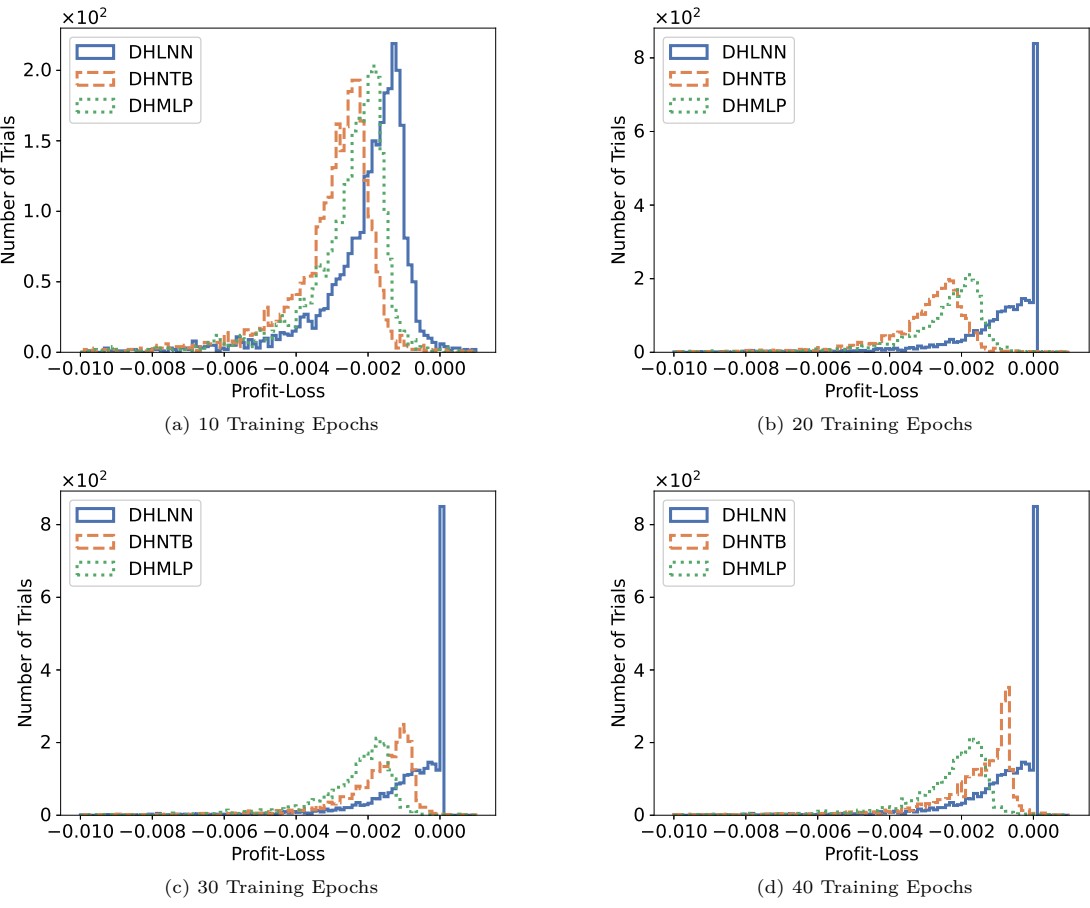

(a) 10 Training Epochs

(b) 20 Training Epochs

(c) 30 Training Epochs

(d) 40 Training Epochs

Figure 9: Comparison of convergence performance across different training epochs $\{10, 20, 30, 40\}$ for deep hedging models applied to a Lookback option with a strike price of 1.0. The analysis considers a transaction cost of $2 \times 10^{-3}$. The purpose is to evaluate the ability of deep hedging models to manage the path dependency inherent in Lookback options and achieve a concentrated PNL distribution around zero, which reflects effective risk management and robust hedging performance.

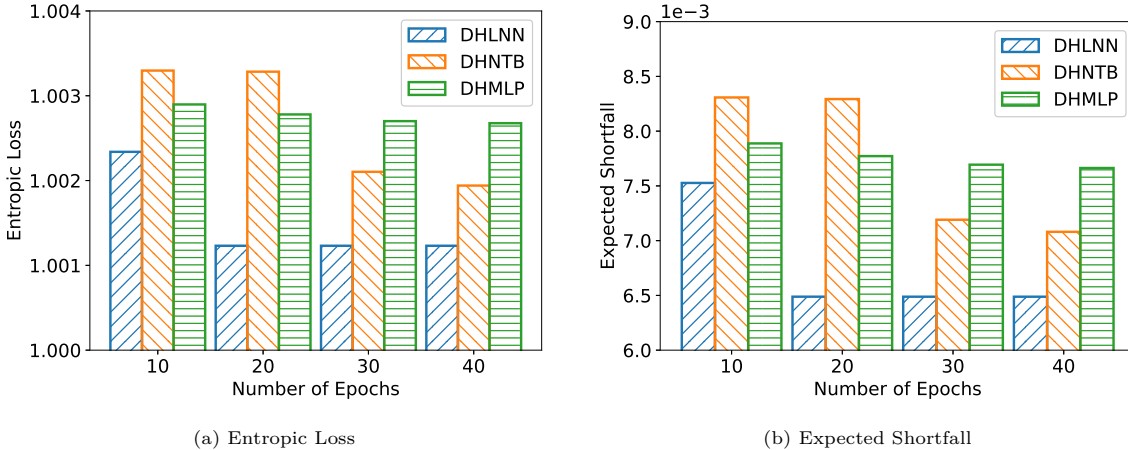

(a) Entropic Loss

(b) Expected Shortfall

Figure 10: Convergence performance comparison over different training epochs $\{10, 20, 30, 40\}$ for deep hedging models applied to a Lookback option with a strike price of 1.0. The analysis evaluates Entropic Loss and Expected Shortfall of the hedging PNL, with a transaction cost of $2 \times 10^{-3}$.

converge faster and more reliably compared to the other methods. In contrast, DHNTB and DHMLP show gradual improvements but continue to lag behind DHLNN, with broader PNL distributions and increased exposure to both over-hedging and under-hedging risks. By 40 epochs as shown in Fig. 9(d), DHLNN achieves near-optimal performance with a highly concentrated PNL distribution around zero. This highlights its robustness and adaptability in managing the path dependency of Lookback options. On the other hand, the baseline methods, particularly DHMLP, struggle to reach a similar level of performance, reflecting their limitations in addressing the complexities of Lookback options.

Fig. 10 provides a detailed analysis of the convergence performance of deep hedging models, including DHLNN, DHNTB, and DHMLP, over a Lookback option with a strike price of 1.0. The evaluation focuses on Entropic Loss and Expected Shortfall, measured over varying training epochs $\{10, 20, 30, 40\}$.

The Entropic Loss results as shown in Fig. 10(a) illustrate that DHLNN achieves consistently lower values compared to the baseline models across all training epochs. At 10 epochs, DHLNN already demonstrates a significant advantage, indicating faster convergence and superior stability during training. As training progresses to 20, 30, and 40 epochs, the Entropic Loss for DHLNN further decreases, reflecting the model's robust ability to mitigate amplified risks and adapt to the complexities of Lookback options. Conversely, DHNTB and DHMLP exhibit slower reductions in Entropic Loss, with values remaining notably higher, highlighting their limited efficiency in stabilizing the hedging process.

The Expected Shortfall results as shown in Fig. 10(b) provide complementary insights into the models' tail risk management capabilities. DHLNN consistently outperforms DHNTB and DHMLP, achieving the lowest Expected Shortfall values across all epochs. This demonstrates the model's superior ability to minimize extreme losses, a critical requirement for effective risk management in real-world financial markets. Furthermore, the gradual improvement in Expected Shortfall over epochs for DHLNN indicates its reliability in converging toward optimal hedging strategies. In contrast, DHNTB and DHMLP show higher Expected Shortfall values throughout the training, underscoring their vulnerability to tail risks and slower convergence rates.

### 5.4.2 Robustness to Market Frictions on Lookback Options with Real Market Data

Fig. 11 illustrates the robustness of various deep hedging models, including DHLNN, DHNTB, and DHMLP, in adapting to increasing transaction costs when applied to Lookback options with a strike price of 1.0. The analysis evaluates the distribution of hedging PNL across four transaction cost levels, i.e., $2 \times 10^{-3}$, $4 \times 10^{-3}$, $6 \times 10^{-3}$, and $8 \times 10^{-3}$, to assess the models' adaptability and efficiency under market frictions.

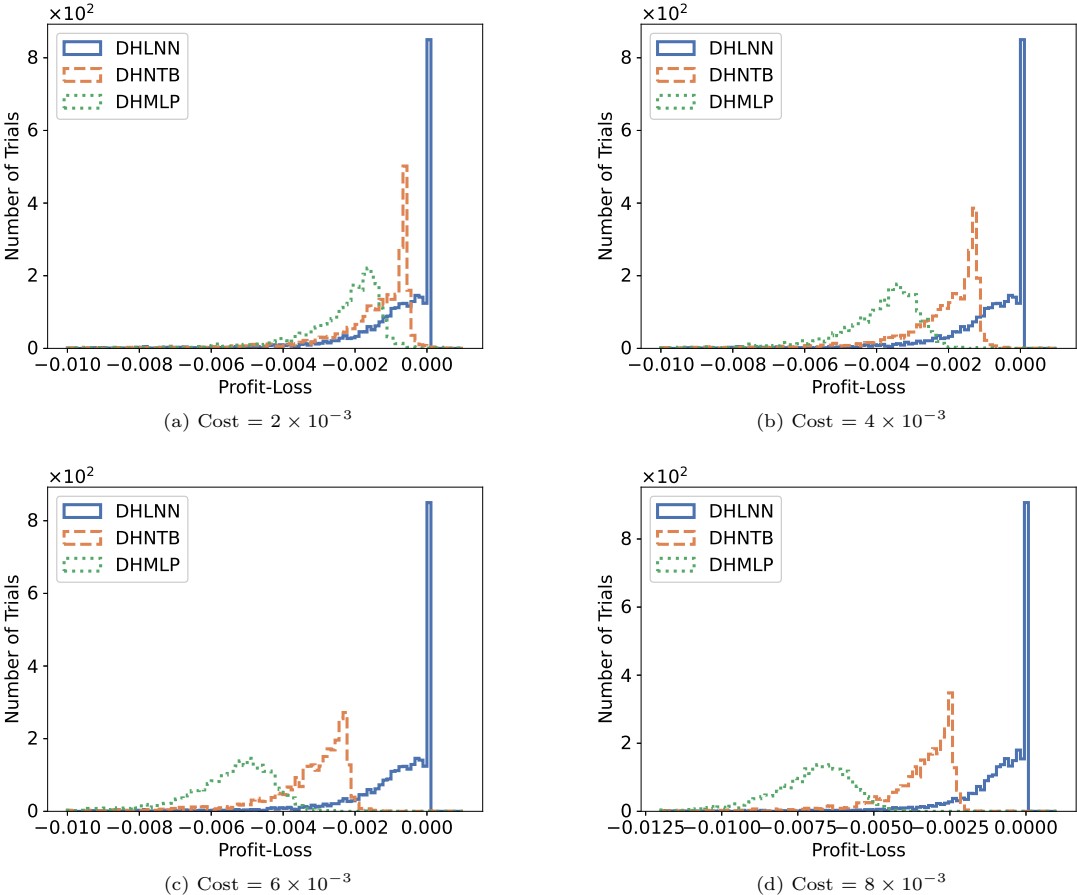

Figure 11: Hedging performance comparison over different transaction costs $\{2 \times 10^{-3}, 4 \times 10^{-3}, 6 \times 10^{-3}, 8 \times 10^{-3}\}$ of the underlying asset for a Lookback option with a strike price of 1.0. The analysis focuses on the distribution of hedging PNL conducted over 50 training epochs.

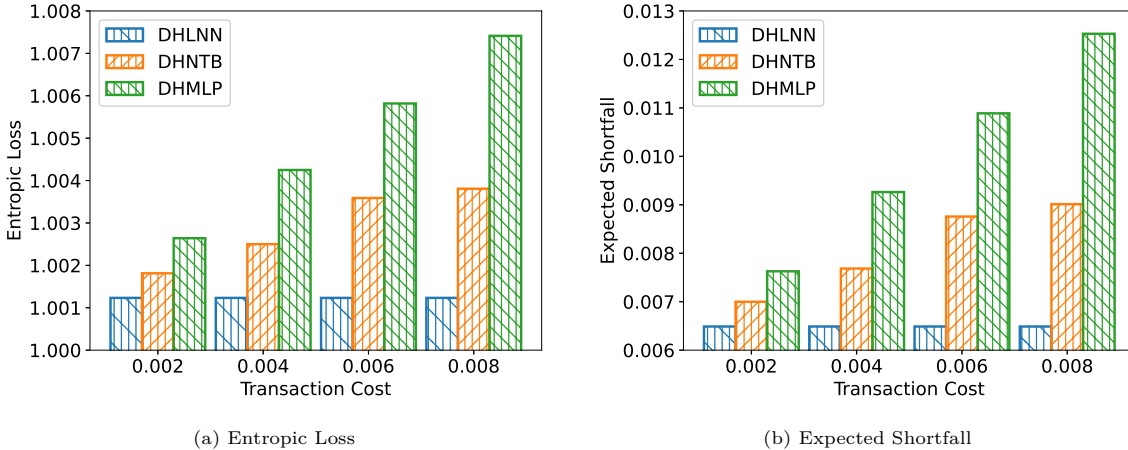

(a) Entropic Loss

(b) Expected Shortfall

Figure 12: Comparison of hedging performance across various transaction costs $\{2 \times 10^{-3}, 4 \times 10^{-3}\}$ for a Lookback option with a strike price of 1.0. The evaluation uses 50 training epochs and focuses on the Entropic Loss and Expected Shortfall of the hedging PNL.

The results show that DHLNN maintains a significantly narrower and more concentrated PNL distribution around zero across all transaction cost levels compared to the baseline methods. At the lowest transaction cost of $2 \times 10^{-3}$ as shown in Fig. 11(a), DHLNN already demonstrates superior performance, with a sharp peak around zero, indicating its ability to achieve a near risk-neutral position effectively. In contrast, both DHNTB and DHMLP exhibit broader PNL distributions, reflecting increased exposure to over-hedging and under-hedging risks.

As the transaction cost increases to $4 \times 10^{-3}$ and $6 \times 10^{-3}$ as shown in Fig. 11(b) and Fig. 11(c), the robustness of DHLNN becomes even more apparent. The model consistently achieves a concentrated PNL distribution, while the baseline methods experience a notable widening of their distributions. This highlights the limitations of DHNTB and DHMLP in managing higher transaction costs, as they are unable to adapt their strategies effectively to mitigate the impact of market frictions.

At the highest transaction cost of $8 \times 10^{-3}$ as shown in Fig. 11(d), DHLNN continues to outperform, maintaining a stable PNL distribution. The baseline methods, on the other hand, show further deterioration in performance, with their PNL distributions becoming increasingly dispersed. This underscores the superior adaptability and efficiency of DHLNN in hedging Lookback options under challenging market conditions.

## 6 Conclusion

This work introduced the DHLNN, a robust framework that addresses computational and practical challenges in deep hedging by integrating periodic fixed-gradient updates, linearized training dynamics, and trajectory-wide optimization. Grounded in financial theory through the Black-Scholes Delta, DHLNN enhances training efficiency, stability, and interpretability, making it a practical and reliable solution for managing derivative liabilities in dynamic markets. Experimental results demonstrated its ability to achieve faster convergence, superior hedging performance, and resilience to noisy data and non-stationarity. Future directions include extending the framework to multi-asset portfolios, incorporating transaction cost models, and integrating adaptive policy optimization via reinforcement learning, paving the way for further innovation in quantitative finance.

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

## A  Weighted Average Price from Order Book

In financial markets, the WAP is a fundamental metric used to assess the value of a tradable asset. Unlike simple measures such as the last traded price, the WAP incorporates both the prices and sizes of orders, offering a more comprehensive and accurate view of the asset's market valuation. The WAP is derived from the order book, which records all outstanding buy (bid) and sell (ask) orders at a given time. By reflecting the cumulative effect of these orders, the WAP provides insights into market depth and liquidity.

We consider a market where time progresses in discrete intervals, represented as

$$t = \{t_0 = 0, t_1, t_2, \cdots, t_n = T\}, \tag{30}$$

where $t_0$ represents the initial time, $t_n$ corresponds to the final time $T$, and $t_i$ denotes an arbitrary time step within this interval. At each time $t_i$, the WAP of the asset, denoted as $P_{t_i}$, is computed by aggregating the weighted average bid and ask prices. The bid prices and sizes are recorded as Bid Price$_j$ and Bid Size$_j$, respectively, where $j$ indexes the set of outstanding buy orders. Similarly, the ask prices and sizes are recorded as Ask Price$_k$ and Ask Size$_k$, where $k$ indexes the set of outstanding sell orders.

The weighted average bid price, denoted as $P_{\text{bid},t_i}$, is calculated by weighting the bid prices according to their respective sizes

$$P_{\text{bid},t_i} = \frac{\sum_j (\text{Bid Price}_j \cdot \text{Bid Size}_j)}{\sum_j \text{Bid Size}_j}. \tag{31}$$

The numerator represents the total value of all bid orders at $t_i$, while the denominator reflects the total bid volume. This ensures that larger orders have a proportionally greater impact on the computed average.

Similarly, the weighted average ask price, $P_{\text{ask},t_i}$, aggregates the ask prices weighted by their respective sizes

$$P_{\text{ask},t_i} = \frac{\sum_k (\text{Ask Price}_k \cdot \text{Ask Size}_k)}{\sum_k \text{Ask Size}_k}. \tag{32}$$

The numerator here accounts for the total value of all ask orders at $t_i$, and the denominator represents the total ask volume, ensuring a fair valuation that reflects market supply.

The WAP at $t_i$ is then computed as the average of the weighted average bid and ask prices

$$P_{t_i} = \frac{P_{\text{bid},t_i} + P_{\text{ask},t_i}}{2}. \tag{33}$$

Over the time horizon $[0, T]$, the series of WAP values forms a time series, expressed as

$$P = \{P_{t_i} \mid P_{t_i} > 0\}_{0 \le t_i \le T}. \tag{34}$$

This series reflects the temporal evolution of the asset's value as determined by market dynamics.

The WAP is particularly valuable because it integrates order size into the pricing mechanism, ensuring that larger orders have a proportionally greater influence on the computed price. This characteristic makes the WAP a more robust and fair measure compared to simplistic metrics. Furthermore, by providing a clearer picture of market liquidity, the WAP helps market participants understand the depth and stability of the market. It also allows traders to anticipate the potential impact of executing large trades, making it an indispensable tool for informed decision-making in financial markets.

# B  Certainty Equivalent of Convex Risk Measures in Indifference Pricing

Indifference pricing provides a robust framework for evaluating derivative prices and hedging strategies under market frictions. It captures the trade-off between minimizing risk and the associated cost of liability management. In this appendix section, we derive the certainty equivalent of convex risk measures in the context of indifference pricing, connecting theoretical constructs to practical hedging.

To achieve indifference between the position $-Z$ and the scenario without liability, which represents the acceptable case with no capital injection, we introduce the indifference price $q(Z)$ as

$$
\begin{aligned}
q(Z) = \inf_{\delta} \; \rho\Big(-Z + \sum_{t_i=0}^{T-1} \delta_{t_i} \Delta P_{t_i} - C_{t_i}|\delta_{t_{i+1}} - \delta_{t_i}|P_{t_i}\Big) \\
- \inf_{\delta} \; \rho\Big(\sum_{t_i=0}^{T-1} \delta_{t_i} \Delta P_{t_i} - C_{t_i}|\delta_{t_{i+1}} - \delta_{t_i}|P_{t_i}\Big).
\end{aligned}
\tag{35}
$$

Indifference pricing provides a natural and versatile framework for characterizing the optimality of hedging strategies and determining the fair price of derivatives in the presence of market frictions. The indifference price $q(Z)$ can be interpreted as the amount of capital needed to injected into the portfolio to eliminate the liability $Z$, which means inject $q(Z)$ can achieve the same expected utility as the case without liability.

The optimal hedging and pricing problem is converted into an optimization problem of the convex risk measure. According to the exponential utility $U(x) = -\exp(-\lambda x)$, we define a continuous, non-decreasing, and convex loss function $L$ as

$$
L(x) = e^{\lambda x} - \frac{1 + \log(\lambda)}{\lambda},
\tag{36}
$$

where $\lambda > 0$ is a risk aversion coefficient. Then, we establish an optimized certainty equivalent of the convex risk measure. This measure can be formulated as an optimization problem, aiming to minimize the certainty equivalent under the given loss function $L$

$$
\rho_Z = \inf_{\theta \in \mathbb{R}} \Big\{\theta + \mathbb{E}[L(Z - \theta - \sum_{t_i=0}^{T-1}(\delta_{t_i}\Delta P_{t_i} - C_{t_i}|\delta_{t_{i+1}} - \delta_{t_i}|P_{t_i}))]\Big\},
\tag{37}
$$

where $\theta$ is an optimization variable. Substituting the loss function $L(x) = e^{\lambda x} - \frac{1+\log(\lambda)}{\lambda}$ into the optimization problem, we get

$$
\rho_Z = \inf_{\theta \in \mathbb{R}} \Big\{\theta + \mathbb{E}[e^{\lambda(Z-\theta-\sum_{t_i=0}^{T-1}(\delta_{t_i}\Delta P_{t_i} - C_{t_i}|\delta_{t_{i+1}} - \delta_{t_i}|P_{t_i}))}] - \frac{1+\log(\lambda)}{\lambda}\Big\}.
\tag{38}
$$

We denote $X$ as follows

$$
X = Z - \sum_{t_i=0}^{T-1}(\delta_{t_i}\Delta P_{t_i} - C_{t_i}|\delta_{t_{i+1}} - \delta_{t_i}|P_{t_i}),
\tag{39}
$$

which simplifies our problem to

$$
\rho_Z = \inf_{\theta \in \mathbb{R}} \Big\{\theta + \mathbb{E}[e^{\lambda(X-\theta)}] - \frac{1+\log(\lambda)}{\lambda}\Big\}.
\tag{40}
$$

We solve the optimization problem by differentiating w.r.t. $\theta$ and setting the derivative to zero to find the optimal $\theta$

$$
\frac{\partial}{\partial \theta}\Big\{\theta + \mathbb{E}[e^{\lambda(X-\theta)}] - \frac{1+\log(\lambda)}{\lambda}\Big\} = 1 - \lambda\mathbb{E}[e^{\lambda(X-\theta)}] = 0.
\tag{41}
$$

Thus, we have

$$
\lambda\mathbb{E}[e^{\lambda(X-\theta)}] = 1 \quad \Rightarrow \quad \mathbb{E}[e^{\lambda(X-\theta)}] = \frac{1}{\lambda}.
\tag{42}
$$

Taking the logarithm on both sides, we get

$$\lambda(X - \theta) = \log\left(\frac{1}{\lambda}\right) = -\log(\lambda) \quad \Rightarrow \quad X - \theta = -\frac{\log(\lambda)}{\lambda}, \tag{43}$$

which leads to

$$\theta = X + \frac{\log(\lambda)}{\lambda}. \tag{44}$$

Substitute the optimal $\theta$ back into the original optimization problem, we obtain the entropic risk measure as

$$\rho_Z = \frac{1}{\lambda} \log(\lambda \mathbb{E}[e^{\lambda(Z - \sum_{t_i=0}^{T-1}(\delta_{t_i}\Delta P_{t_i} - C_{t_i}|\delta_{t_{i+1}} - \delta_{t_i}|P_{t_i}))}]). \tag{45}$$

This completes the detailed derivation of converting the optimal hedging and pricing problem into an optimization problem using a convex risk measure with an exponential utility function.

The fair price $q(Z)$ corresponds to the cash amount that renders a hedger the equivalent situation of settling a liability with $q(Z)$ and having no liability, assuming optimal hedging. The optimal convex risk measure is construed as the residual risk of the derivative post-optimal hedging. Building on the foundation of the optimal hedge within the framework of the hedger's risk measure, the hedger quotes a price that compensates for the remaining risk after hedging.

## C  Anchor Hedge Strategy Using Black-Scholes Delta

The anchor hedge strategy involves using the Black-Scholes delta, which is a derivative of the Black-Scholes formula commonly used in financial mathematics to hedge options. This strategy leverages the sensitivity of the option's price to small changes in the price of the underlying asset, known as the delta, to create a hedged position. We present a detailed explanation of the components and the process in the following.

The variable $bs_{t_i}$ is defined as

$$bs_{t_i} = \frac{\log(\frac{P_{t_i}}{P_s}) + \frac{\sigma^2(T-t_i)}{2}}{\sigma\sqrt{T - t_i}} \tag{46}$$

where $P_{t_i}$ is the current price of the underlying asset at time $t_i$, $P_s$ is the strike price of the option, $\sigma$ is the volatility of the underlying asset's returns, $T$ is the time to maturity of the option, and $t_i$ is the current time. This equation calculates the standardized distance between the logarithm of the current price and the strike price, adjusted for the time remaining until maturity and the volatility. It essentially measures how far the current price is from the strike price in standard deviation units, taking into account the time decay.

The Black-Scholes delta $\delta_{t_i}^{bs}$ is given by

$$\delta_{t_i}^{bs} = \frac{1}{2}\left(1 + \frac{2}{\sqrt{\pi}}\int_0^{bs_{t_i}} e^{-t^2} dt\right) \tag{47}$$

which represents the hedge ratio, or the sensitivity of the option's price to small changes in the price of the underlying asset. The delta is the probability that the option will end up in-the-money, which means it will have a positive payoff at expiration. The term $\frac{2}{\sqrt{\pi}}\int_0^{bs_{t_i}} e^{-t^2} dt$ is the error function $\mathrm{erf}(bs_{t_i})$, related to the normal distribution and helps in computing the probabilities associated with the normal distribution.

The delta $\delta_{t_i}^{bs}$ ranges from 0 to 1. When the underlying asset price $P_{t_i}$ is much higher than the strike price $P_s$, $bs_{t_i}$ will be large, and $\delta_{t_i}^{bs}$ will approach 1, indicating a high probability that the option will end up in-the-money. Conversely, if $P_{t_i}$ is much lower than $P_s$, $bs_{t_i}$ will be negative and large in magnitude, and $\delta_{t_i}^{bs}$ will approach 0, indicating a low probability of the option being in-the-money.

The anchor hedge strategy involves using this delta to hedge an option position. By holding $\delta_{t_i}^{bs}$ units of the underlying asset for each option, the trader can create a hedged position that mitigates the risk from small movements in the underlying asset's price. The delta is recalculated periodically as $P_{t_i}$ and $t_i$ change, to maintain the hedge.

# D  Analysis of Approximation Quality in Hedging Model

In the main text, we introduced the hedging model using a streamlined notation $f(\boldsymbol{x}_{t_i}, \boldsymbol{w})$ to emphasize the high-level functionality of the neural network: mapping input market information $\boldsymbol{x}_{t_i}$ to the permissible band for hedging positions. To maintain clarity for a broader audience, this notation abstracts the underlying mathematical complexity of the neural network's computations.

In this section, we present the detailed formulation of the model as $f_k(\boldsymbol{x}_{t_i}, \boldsymbol{w}, \boldsymbol{a}_k)$, which explicitly captures the structure and operations within the network. This formulation includes the parameters $\boldsymbol{w}$ representing weights and biases of the hidden layers and $\boldsymbol{a}_k$ representing weights associated with the outputs and describes how the input $\boldsymbol{x}_{t_i}$ is transformed through successive layers of the network to produce the outputs $f_1(\boldsymbol{x}_{t_i})$ and $f_2(\boldsymbol{x}_{t_i})$. These outputs define the permissible hedging range, addressing position-dependence effectively.

## D.1  Model Function Formulation

The neural network processes market information as inputs $\boldsymbol{x}_{t_i}$ and produces a permissible band for the subsequent position, effectively addressing the issue of position-dependence. The parameters of the neural network model are represented by the weight matrices and bias vectors across multiple layers. Let $\boldsymbol{W}^{(l)}$ and $\boldsymbol{b}^{(l)}$ denote the weights and biases of the $l$-th layer, where $l = 1, \ldots, L-1$. The hidden layers are indexed by $l = 1, \ldots, L-1$. The parameters of the neural network model are represented by $\boldsymbol{w} = \{\boldsymbol{W}^{(l)}, \boldsymbol{b}^{(l)}\}_{l=1}^{L-1}$ and $\boldsymbol{a_1}, \boldsymbol{a_2} \in \mathbb{R}^m$.

Given an input $\boldsymbol{x}_{t_i}$, the neural network processes this input through $L-1$ hidden layers. The transformation starts with the first hidden layer, which computes

$$\boldsymbol{h}^{(1)} = \sigma(\boldsymbol{W}^{(1)} \boldsymbol{x}_{t_i} + \boldsymbol{b}^{(1)}). \tag{48}$$

Subsequent hidden layers apply the transformation

$$\boldsymbol{h}^{(l)} = \sigma(\boldsymbol{W}^{(l)} \boldsymbol{h}^{(l-1)} + \boldsymbol{b}^{(l)}) \quad \text{for} \quad l = 2, \ldots, L-1. \tag{49}$$

Finally, the output layer applies a linear transformation to the outputs of the last hidden layer. This transformation is weighted by the vectors $\boldsymbol{a}_1$ and $\boldsymbol{a}_2$ to produce the two outputs $f_1(\boldsymbol{x}_{t_i})$ and $f_2(\boldsymbol{x}_{t_i})$. The model function, denoted as $f : \mathbb{R}^d \to \mathbb{R}^2$ with $L$ hidden layers comprising is defined as

$$\left\{ \boldsymbol{f}_k(\boldsymbol{x}_{t_i}, \boldsymbol{w}, \boldsymbol{a}_k) = \frac{1}{\sqrt{m}} \sum_{j=1}^m \boldsymbol{a}_{k,j} \prod_{l=1}^L \left[ \sigma(\boldsymbol{W}^{(l)} \boldsymbol{h}^{(l-1)} + \boldsymbol{b}^{(l)}) \right] \right\}_{k=1,2}. \tag{50}$$

The neural network representation facilitates the translation of the hedging problem into the optimization of a set of parameters $\boldsymbol{w} = \{\boldsymbol{W}^{(l)}, \boldsymbol{b}^{(l)}\}_{l=1}^{L-1}$ of the neural network, guided by the convex risk measure in (45). The minimizer of (45) provides an approximate value for the indifference price, as demonstrated in (35). This comprehensive methodology offers a promising avenue for addressing challenges in neural network hedging within the context of financial markets.

The neural network's design effectively tackles the challenge of position-dependence, which refers to the difficulty of making predictions or decisions that are influenced by the specific position or state of a financial instrument. By using this approach, the model is able to incorporate various market factors and dependencies into its predictions, providing a flexible and adaptive mechanism for financial decision-making. This capability is crucial in financial contexts where market conditions and positions can vary widely, ensuring that the network's outputs remain relevant and accurate across different scenarios.

To simplify the analysis, we assume each variable in $\{\boldsymbol{a}_k\}_{k=1,2}$ is sampled uniformly at random and exclude it from the training procedure, focusing solely on the weight matrices $\boldsymbol{W}^{(l)}$ and bias vectors $\boldsymbol{b}^{(l)}$, which make up the variable $\boldsymbol{w} = \{\boldsymbol{W}^{(l)}, \boldsymbol{b}^{(l)}\}_{l=1}^{L-1}$. We consider the following assumptions, i.e., the derivative of the activation function $|\sigma'(z)|$ is bounded by $M_1$ for all $z$, the input vectors $\boldsymbol{h}^{(l-1)}$ are bounded such that $\|\boldsymbol{h}^{(l-1)}\| \leq H$, and the elements $\boldsymbol{a}_{k,j}$ are constants with a maximum absolute value $A = \max_j |\boldsymbol{a}_{k,j}|$.

## D.2 Gradient Magnitude Analysis

To compute the gradient of the model function w.r.t. the weight matrices $\boldsymbol{W}^{(l)}$ and bias vectors $\boldsymbol{b}^{(l)}$, we use the chain rule, where the terms $\prod_{i=1}^{l-1} \sigma(\boldsymbol{W}^{(i)}\boldsymbol{h}^{(i-1)} + \boldsymbol{b}^{(i)})$ capture the propagation of the gradient through all previous layers. The gradient w.r.t. the weight matrices $\boldsymbol{W}^{(l)}$ is derived as

$$\nabla_{\boldsymbol{W}^{(l)}} f_k(\boldsymbol{x}_{t_i}, \boldsymbol{w}, \boldsymbol{a}_k) = \nabla_{\boldsymbol{h}^{(l)}} f_k(\boldsymbol{x}_{t_i}, \boldsymbol{w}, \boldsymbol{a}_k) \cdot \nabla_{\boldsymbol{W}^{(l)}} \boldsymbol{h}^{(l)}, \tag{51}$$

where

$$\nabla_{\boldsymbol{W}^{(l)}} \boldsymbol{h}^{(l)} = \sigma'(\boldsymbol{W}^{(l)}\boldsymbol{h}^{(l-1)} + \boldsymbol{b}^{(l)}) \cdot \boldsymbol{h}^{(l-1)} \tag{52}$$

Combining (51) and (52), we have

$$\nabla_{\boldsymbol{W}^{(l)}} f_k(\boldsymbol{x}_{t_i}, \boldsymbol{w}, \boldsymbol{a}_k) = \frac{1}{\sqrt{m}} \sum_{j=1}^{m} \boldsymbol{a}_{k,j} \left( \prod_{i=1}^{l-1} \sigma(\boldsymbol{W}^{(i)}\boldsymbol{h}^{(i-1)} + \boldsymbol{b}^{(i)}) \right) \cdot \sigma'(\boldsymbol{W}^{(l)}\boldsymbol{h}^{(l-1)} + \boldsymbol{b}^{(l)}) \cdot \boldsymbol{h}^{(l-1)}. \tag{53}$$

The gradient w.r.t. the bias vectors $\boldsymbol{b}^{(l)}$ is derived as

$$\nabla_{\boldsymbol{b}^{(l)}} f_k(\boldsymbol{x}_{t_i}, \boldsymbol{w}, \boldsymbol{a}_k) = \nabla_{\boldsymbol{h}^{(l)}} f_k(\boldsymbol{x}_{t_i}, \boldsymbol{w}, \boldsymbol{a}_k) \cdot \nabla_{\boldsymbol{b}^{(l)}} \boldsymbol{h}^{(l)}, \tag{54}$$

where

$$\nabla_{\boldsymbol{b}^{(l)}} \boldsymbol{h}^{(l)} = \sigma'(\boldsymbol{W}^{(l)}\boldsymbol{h}^{(l-1)} + \boldsymbol{b}^{(l)}). \tag{55}$$

Combining (54) and (55), we have

$$\nabla_{\boldsymbol{b}^{(l)}} f_k(\boldsymbol{x}_{t_i}, \boldsymbol{w}, \boldsymbol{a}_k) = \frac{1}{\sqrt{m}} \sum_{j=1}^{m} \boldsymbol{a}_{k,j} \left( \prod_{i=1}^{l-1} \sigma(\boldsymbol{W}^{(i)}\boldsymbol{h}^{(i-1)} + \boldsymbol{b}^{(i)}) \right) \cdot \sigma'(\boldsymbol{W}^{(l)}\boldsymbol{h}^{(l-1)} + \boldsymbol{b}^{(l)}). \tag{56}$$

The product term $\prod_{i=1}^{l-1} \sigma(\boldsymbol{W}^{(i)}\boldsymbol{h}^{(i-1)} + \boldsymbol{b}^{(i)})$ can be bounded by $M_1^{l-1}$. Next, consider $\sigma'(\boldsymbol{W}^{(l)}\boldsymbol{h}^{(l-1)} + \boldsymbol{b}^{(l)}) \cdot \boldsymbol{h}^{(l-1)}$. Using the bounds $|\sigma'(\boldsymbol{W}^{(l)}\boldsymbol{h}^{(l-1)} + \boldsymbol{b}^{(l)})| \leq M_1$ and $\|\boldsymbol{h}^{(l-1)}\| \leq H$, we get

$$|\sigma'(\boldsymbol{W}^{(l)}\boldsymbol{h}^{(l-1)} + \boldsymbol{b}^{(l)}) \cdot \boldsymbol{h}^{(l-1)}| \leq M_1 H. \tag{57}$$

Then, considering the sum over $m$ terms, we have

$$\left\| \frac{1}{\sqrt{m}} \sum_{j=1}^{m} \boldsymbol{a}_{k,j} \left( \prod_{i=1}^{l-1} \sigma(\boldsymbol{W}^{(i)}\boldsymbol{h}^{(i-1)} + \boldsymbol{b}^{(i)}) \right) \cdot \sigma'(\boldsymbol{W}^{(l)}\boldsymbol{h}^{(l-1)} + \boldsymbol{b}^{(l)}) \cdot \boldsymbol{h}^{(l-1)} \right\| \leq \frac{1}{\sqrt{m}} \sum_{j=1}^{m} |\boldsymbol{a}_{k,j}| M_1^{l-1} M_1 H. \tag{58}$$

Since $\boldsymbol{a}_{k,j}$ are constants and $A$ is the maximum of their absolute values

$$\frac{1}{\sqrt{m}} \sum_{j=1}^{m} |\boldsymbol{a}_{k,j}| \leq A\sqrt{m}. \tag{59}$$

Combining these bounds, we get

$$\|\nabla_{\boldsymbol{W}^{(l)}} f_k(\boldsymbol{x}_{t_i}, \boldsymbol{w}, \boldsymbol{a}_k)\| \leq \frac{1}{\sqrt{m}} \cdot A\sqrt{m} \cdot M_1^l \cdot H = AM_1^l H. \tag{60}$$

Similarly, for the bias vectors, given the gradient expression

$$\nabla_{\boldsymbol{b}^{(l)}} f_k(\boldsymbol{x}_{t_i}, \boldsymbol{w}, \boldsymbol{a}_k) = \frac{1}{\sqrt{m}} \sum_{j=1}^{m} \boldsymbol{a}_{k,j} \left( \prod_{i=1}^{l-1} \sigma(\boldsymbol{W}^{(i)}\boldsymbol{h}^{(i-1)} + \boldsymbol{b}^{(i)}) \right) \cdot \sigma'(\boldsymbol{W}^{(l)}\boldsymbol{h}^{(l-1)} + \boldsymbol{b}^{(l)}). \tag{61}$$

we can follow a similar analysis. We have

$$\left\| \frac{1}{\sqrt{m}} \sum_{j=1}^{m} \boldsymbol{a}_{k,j} \left( \prod_{i=1}^{l-1} \sigma(\boldsymbol{W}^{(i)} \boldsymbol{h}^{(i-1)} + \boldsymbol{b}^{(i)}) \right) \cdot \sigma'(\boldsymbol{W}^{(l)} \boldsymbol{h}^{(l-1)} + \boldsymbol{b}^{(l)}) \right\| \leq \frac{1}{\sqrt{m}} \sum_{j=1}^{m} |\boldsymbol{a}_{k,j}| M_1^l. \tag{62}$$

Using the same argument for $|\boldsymbol{a}_{k,j}|$ in (59), we get

$$\|\nabla_{\boldsymbol{b}^{(l)}} f_k(\boldsymbol{x}_{t_i}, \boldsymbol{w}, \boldsymbol{a}_k)\| \leq \frac{1}{\sqrt{m}} \cdot A\sqrt{m} \cdot M_1^l = AM_1^l. \tag{63}$$

The bound of the gradient for both the weight matrices and the bias vectors can be summarized as

$$\|\nabla_{\boldsymbol{W}^{(l)}} f_k(\boldsymbol{x}_{t_i}, \boldsymbol{w}, \boldsymbol{a}_k)\| \leq AM_1^l H, \tag{64}$$

and

$$\|\nabla_{\boldsymbol{b}^{(l)}} f_k(\boldsymbol{x}_{t_i}, \boldsymbol{w}, \boldsymbol{a}_k)\| \leq AM_1^l. \tag{65}$$

Combining the effects from both the weights and biases, we can write the overall bound on the gradient w.r.t. the model parameters $\boldsymbol{w}$ as

$$\|\nabla_{\boldsymbol{w}} f_k(\boldsymbol{x}_{t_i}, \boldsymbol{w}, \boldsymbol{a}_k)\| \leq AM_1^l(H + 1). \tag{66}$$

The analysis above offers essential insights into the gradient behavior within the hedging model's training process. By applying the chain rule, it is evident that the gradient magnitudes $\|\nabla_{\boldsymbol{w}} f_k(\boldsymbol{x}_{t_i}, \boldsymbol{w}, \boldsymbol{a}_k)\|$ are exponentially influenced by the network's depth $l$. This highlights the importance of selecting appropriate initializations and activation functions. When the magnitude of the activation function's gradient $M_1$ is controlled around 1, the gradient magnitudes $\|\nabla_{\boldsymbol{w}} f_k(\boldsymbol{x}_{t_i}, \boldsymbol{w}, \boldsymbol{a}_k)\|$ tend to stabilize, converging to a constant value as the network depth increases. This behavior strongly supports the subsequent analysis, demonstrating that our approximation maintains high quality, closely aligning with the linearized hedging model.

### D.3 Gradient Changing Rate Analysis

To analyze the gradient changing rate, we focus on the rate at which the gradient of the loss function changes w.r.t. the model parameters. This involves examining the difference in gradients between two different iterations or parameter settings. We start with the gradient of the model function $f_k(\boldsymbol{x}_{t_i}, \boldsymbol{w}, \boldsymbol{a}_k)$ w.r.t. the weight matrices $\boldsymbol{W}^{(l)}$ and bias vectors $\boldsymbol{b}^{(l)}$. To measure the gradient changing rate, we consider the difference between the gradients at two different parameter settings, $\boldsymbol{w}$ and $\boldsymbol{w}'$.

For the weight matrices, the gradient difference is

$$\Delta\nabla_{\boldsymbol{W}^{(l)}} f_k = \nabla_{\boldsymbol{W}^{(l)}} f_k(\boldsymbol{x}_{t_i}, \boldsymbol{w}, \boldsymbol{a}_k) - \nabla_{\boldsymbol{W}^{(l)}} f_k(\boldsymbol{x}_{t_i}, \boldsymbol{w}', \boldsymbol{a}_k). \tag{67}$$

Substituting the gradient expressions, we get

$$\begin{aligned}
\Delta\nabla_{\boldsymbol{W}^{(l)}} f_k = \frac{1}{\sqrt{m}} \sum_{j=1}^{m} \boldsymbol{a}_{k,j} &\left( \left( \prod_{i=1}^{l-1} \sigma(\boldsymbol{W}^{(i)} \boldsymbol{h}^{(i-1)} + \boldsymbol{b}^{(i)}) \right) \cdot \sigma'(\boldsymbol{W}^{(l)} \boldsymbol{h}^{(l-1)} + \boldsymbol{b}^{(l)}) \cdot \boldsymbol{h}^{(l-1)} \right. \\
&\left. - \left( \prod_{i=1}^{l-1} \sigma(\boldsymbol{W}'^{(i)} \boldsymbol{h}'^{(i-1)} + \boldsymbol{b}'^{(i)}) \right) \cdot \sigma'(\boldsymbol{W}'^{(l)} \boldsymbol{h}'^{(l-1)} + \boldsymbol{b}'^{(l)}) \cdot \boldsymbol{h}'^{(l-1)} \right).
\end{aligned} \tag{68}$$

Using the triangle inequality, we can bound the difference as

$$\begin{aligned}
\|\Delta\nabla_{\boldsymbol{W}^{(l)}} f_k\| \leq \frac{1}{\sqrt{m}} \sum_{j=1}^{m} |\boldsymbol{a}_{k,j}| &\left\| \left( \prod_{i-1}^{l-1} \sigma(\boldsymbol{W}^{(i)} \boldsymbol{h}^{(i-1)} + \boldsymbol{b}^{(i)}) \right) \cdot \sigma'(\boldsymbol{W}^{(l)} \boldsymbol{h}^{(l-1)} + \boldsymbol{b}^{(l)}) \cdot \boldsymbol{h}^{(l-1)} \right. \\
&\left. - \left( \prod_{i=1}^{l-1} \sigma(\boldsymbol{W}'^{(i)} \boldsymbol{h}'^{(i-1)} + \boldsymbol{b}'^{(i)}) \right) \cdot \sigma'(\boldsymbol{W}'^{(l)} \boldsymbol{h}'^{(l-1)} + \boldsymbol{b}'^{(l)}) \cdot \boldsymbol{h}'^{(l-1)} \right\|.
\end{aligned} \tag{69}$$

Similarly, for the bias vectors, the gradient difference is

$$\Delta \nabla_{\boldsymbol{b}^{(l)}} f_k = \nabla_{\boldsymbol{b}^{(l)}} f_k(\boldsymbol{x}_{t_i}, \boldsymbol{w}, \boldsymbol{a}_k) - \nabla_{\boldsymbol{b}^{(l)}} f_k(\boldsymbol{x}_{t_i}, \boldsymbol{w}', \boldsymbol{a}_k). \tag{70}$$

Substituting the gradient expressions, we get

$$\Delta \nabla_{\boldsymbol{b}^{(l)}} f_k = \frac{1}{\sqrt{m}} \sum_{j=1}^{m} \boldsymbol{a}_{k,j} \left( \left( \prod_{i=1}^{l-1} \sigma(\boldsymbol{W}^{(i)} \boldsymbol{h}^{(i-1)} + \boldsymbol{b}^{(i)}) \right) \cdot \sigma'(\boldsymbol{W}^{(l)} \boldsymbol{h}^{(l-1)} + \boldsymbol{b}^{(l)}) \right.$$
$$\left. - \left( \prod_{i=1}^{l-1} \sigma(\boldsymbol{W}'^{(i)} \boldsymbol{h}'^{(i-1)} + \boldsymbol{b}'^{(i)}) \right) \cdot \sigma'(\boldsymbol{W}'^{(l)} \boldsymbol{h}'^{(l-1)} + \boldsymbol{b}'^{(l)}) \right). \tag{71}$$

Using the triangle inequality, we can bound the difference as

$$\|\Delta \nabla_{\boldsymbol{b}^{(l)}} f_k\| \leq \frac{1}{\sqrt{m}} \sum_{j=1}^{m} |\boldsymbol{a}_{k,j}| \left\| \left( \prod_{i=1}^{l-1} \sigma(\boldsymbol{W}^{(i)} \boldsymbol{h}^{(i-1)} + \boldsymbol{b}^{(i)}) \right) \cdot \sigma'(\boldsymbol{W}^{(l)} \boldsymbol{h}^{(l-1)} + \boldsymbol{b}^{(l)}) \right.$$
$$\left. - \left( \prod_{i=1}^{l-1} \sigma(\boldsymbol{W}'^{(i)} \boldsymbol{h}'^{(i-1)} + \boldsymbol{b}'^{(i)}) \right) \cdot \sigma'(\boldsymbol{W}'^{(l)} \boldsymbol{h}'^{(l-1)} + \boldsymbol{b}'^{(l)}) \right\|. \tag{72}$$

Let $\Delta \boldsymbol{W}^{(l)} = \boldsymbol{W}^{(l)} - \boldsymbol{W}'^{(l)}$ and $\Delta \boldsymbol{b}^{(l)} = \boldsymbol{b}^{(l)} - \boldsymbol{b}'^{(l)}$. Given the bounds $|\sigma'(\boldsymbol{W}^{(l)} \boldsymbol{h}^{(l-1)} + \boldsymbol{b}^{(l)})| \leq M_1$, $\|\boldsymbol{h}^{(l-1)}\| \leq H$, and $|\boldsymbol{a}_{k,j}| \leq A$, we can derive

$$\|\Delta \nabla_{\boldsymbol{W}^{(l)}} f_k\| \leq \frac{1}{\sqrt{m}} \sum_{j=1}^{m} A M_1^{l-1} M_1 H \|\Delta \boldsymbol{W}^{(l)}\| = A M_1^l H \|\Delta \boldsymbol{W}^{(l)}\|, \tag{73}$$

and

$$\|\Delta \nabla_{\boldsymbol{b}^{(l)}} f_k\| \leq \frac{1}{\sqrt{m}} \sum_{j=1}^{m} A M_1^{l-1} M_1 \|\Delta \boldsymbol{b}^{(l)}\| = A M_1^l \|\Delta \boldsymbol{b}^{(l)}\|. \tag{74}$$

To include the Hessian matrix in our analysis, we use the Taylor expansion around the parameter setting $\boldsymbol{w}$

$$\nabla_{\boldsymbol{W}^{(l)}} f_k(\boldsymbol{w}') \approx \nabla_{\boldsymbol{W}^{(l)}} f_k(\boldsymbol{w}) + \nabla_{\boldsymbol{W}^{(l)}}^2 f_k(\boldsymbol{w}) \cdot (\boldsymbol{W}'^{(l)} - \boldsymbol{W}^{(l)}), \tag{75}$$

where $\nabla_{\boldsymbol{W}^{(l)}}^2 f_k(\boldsymbol{w})$ denotes the Hessian matrix of the loss function w.r.t. $\boldsymbol{W}^{(l)}$ at $\boldsymbol{w}$. Therefore, the gradient difference is

$$\Delta \nabla_{\boldsymbol{W}^{(l)}} f_k = \nabla_{\boldsymbol{W}^{(l)}}^2 f_k(\boldsymbol{w}) \cdot (\boldsymbol{W}'^{(l)} - \boldsymbol{W}^{(l)}). \tag{76}$$

The norm of this gradient difference is

$$\|\Delta \nabla_{\boldsymbol{W}^{(l)}} f_k\| = \left\| \nabla_{\boldsymbol{W}^{(l)}}^2 f_k(\boldsymbol{w}) \cdot (\boldsymbol{W}'^{(l)} - \boldsymbol{W}^{(l)}) \right\|. \tag{77}$$

Similarly, for the bias vectors, we have

$$\Delta \nabla_{\boldsymbol{b}^{(l)}} f_k = \nabla_{\boldsymbol{b}^{(l)}}^2 f_k(\boldsymbol{w}) \cdot (\boldsymbol{b}'^{(l)} - \boldsymbol{b}^{(l)}). \tag{78}$$

The norm of the gradient difference for the bias vectors is

$$\|\Delta \nabla_{\boldsymbol{b}^{(l)}} f_k\| = \left\| \nabla_{\boldsymbol{b}^{(l)}}^2 f_k(\boldsymbol{w}) \cdot (\boldsymbol{b}'^{(l)} - \boldsymbol{b}^{(l)}) \right\|. \tag{79}$$

Let $\Delta \boldsymbol{W}^{(l)} = \boldsymbol{W}^{(l)} - \boldsymbol{W}'^{(l)}$ and $\Delta \boldsymbol{b}^{(l)} = \boldsymbol{b}^{(l)} - \boldsymbol{b}'^{(l)}$, and given the bounds on the Hessian, we can derive

$$\|\Delta \nabla_{\boldsymbol{W}^{(l)}} f_k\| \leq \|\nabla_{\boldsymbol{W}^{(l)}}^2 f_k(\boldsymbol{w})\| \cdot \|\Delta \boldsymbol{W}^{(l)}\|, \tag{80}$$

and

$$\|\Delta \nabla_{\boldsymbol{b}^{(l)}} f_k\| \leq \|\nabla_{\boldsymbol{b}^{(l)}}^2 f_k(\boldsymbol{w})\| \cdot \|\Delta \boldsymbol{b}^{(l)}\|. \tag{81}$$

### D.4 Hessian Bound Analysis

To bound the Hessian of the model function $f_k(\boldsymbol{x}, \boldsymbol{w}, \boldsymbol{a}_k)$ w.r.t. the weight matrices $\boldsymbol{W}^{(l)}$ and bias vectors $\boldsymbol{b}^{(l)}$, we need to consider the maximum values that the second-order partial derivatives can attain.

First, for the weight matrices $\boldsymbol{W}^{(l)}$, the second-order partial derivatives are given by

$$\nabla^2_{\boldsymbol{W}^{(l)}} f_k(\boldsymbol{x}, \boldsymbol{w}, \boldsymbol{a}_k) = \frac{1}{\sqrt{m}} \sum_{j=1}^{m} \boldsymbol{a}_{k,j} \prod_{l=1}^{L} \left[ \sigma''(\boldsymbol{W}^{(l)} \boldsymbol{h}^{(l-1)} + \boldsymbol{b}^{(l)}) \boldsymbol{h}^{(l-1)} (\boldsymbol{h}^{(l-1)})^\top + \sigma'(\boldsymbol{W}^{(l)} \boldsymbol{h}^{(l-1)} + \boldsymbol{b}^{(l)}) \right]. \quad (82)$$

For the bias vectors $\boldsymbol{b}^{(l)}$, the second-order partial derivatives are

$$\nabla^2_{\boldsymbol{b}^{(l)}} f_k(\boldsymbol{x}, \boldsymbol{w}, \boldsymbol{a}_k) = \frac{1}{\sqrt{m}} \sum_{j=1}^{m} \boldsymbol{a}_{k,j} \prod_{l=1}^{L} \left[ \sigma''(\boldsymbol{W}^{(l)} \boldsymbol{h}^{(l-1)} + \boldsymbol{b}^{(l)}) \right]. \quad (83)$$

By making another assumption that the activation function $\sigma$ and its derivatives $\sigma''$ are bounded, where $|\sigma''(z)| \leq M_2$ for all $z$. The norm of each term in the product can be bounded by

$$\left| \sigma''(\boldsymbol{W}^{(l)} \boldsymbol{h}^{(l-1)} + \boldsymbol{b}^{(l)}) \boldsymbol{h}^{(l-1)} (\boldsymbol{h}^{(l-1)})^\top \right| \leq M_2 H^2 \quad (84)$$

and

$$\left| \sigma'(\boldsymbol{W}^{(l)} \boldsymbol{h}^{(l-1)} + \boldsymbol{b}^{(l)}) \right| \leq M_1. \quad (85)$$

Thus, the norm of the product term is

$$\left| \prod_{l=1}^{L} \left[ \sigma''(\boldsymbol{W}^{(l)} \boldsymbol{h}^{(l-1)} + \boldsymbol{b}^{(l)}) \boldsymbol{h}^{(l-1)} (\boldsymbol{h}^{(l-1)})^\top + \sigma'(\boldsymbol{W}^{(l)} \boldsymbol{h}^{(l-1)} + \boldsymbol{b}^{(l)}) \right] \right| \leq (M_2 H^2 + M_1)^L. \quad (86)$$

Since $\boldsymbol{a}_{k,j}$ are constants, the sum is bounded by

$$\left| \frac{1}{\sqrt{m}} \sum_{j=1}^{m} \boldsymbol{a}_{k,j} \prod_{l=1}^{L} \left[ \sigma''(\boldsymbol{W}^{(l)} \boldsymbol{h}^{(l-1)} + \boldsymbol{b}^{(l)}) \boldsymbol{h}^{(l-1)} (\boldsymbol{h}^{(l-1)})^\top + \sigma'(\boldsymbol{W}^{(l)} \boldsymbol{h}^{(l-1)} + \boldsymbol{b}^{(l)}) \right] \right|$$
$$\leq \frac{1}{\sqrt{m}} \sum_{j=1}^{m} |\boldsymbol{a}_{k,j}| (M_2 H^2 + M_1)^L. \quad (87)$$

Letting $A = \max_j |\boldsymbol{a}_{k,j}|$, we get

$$\left| \nabla^2_{\boldsymbol{W}^{(l)}} f_k \right| \leq \frac{1}{\sqrt{m}} m A (M_2 H^2 + M_1)^L = A (M_2 H^2 + M_1)^L. \quad (88)$$

Next, consider the Hessian for the bias vectors. Using the bound on $\sigma''$, we have

$$\left| \prod_{l=1}^{L} \sigma''(\boldsymbol{W}^{(l)} \boldsymbol{h}^{(l-1)} + \boldsymbol{b}^{(l)}) \right| \leq M_2^L. \quad (89)$$

The sum is bounded by

$$\left| \frac{1}{\sqrt{m}} \sum_{j=1}^{m} \boldsymbol{a}_{k,j} \prod_{l=1}^{L} \sigma''(\boldsymbol{W}^{(l)} \boldsymbol{h}^{(l-1)} + \boldsymbol{b}^{(l)}) \right| \leq \frac{1}{\sqrt{m}} \sum_{j=1}^{m} |\boldsymbol{a}_{k,j}| M_2^L. \quad (90)$$

Combining with $A = \max_j |\boldsymbol{a}_{k,j}|$, we get

$$\left| \nabla^2_{\boldsymbol{b}^{(l)}} f_k \right| \leq \frac{1}{\sqrt{m}} m A M_2^L = A M_2^L. \quad (91)$$

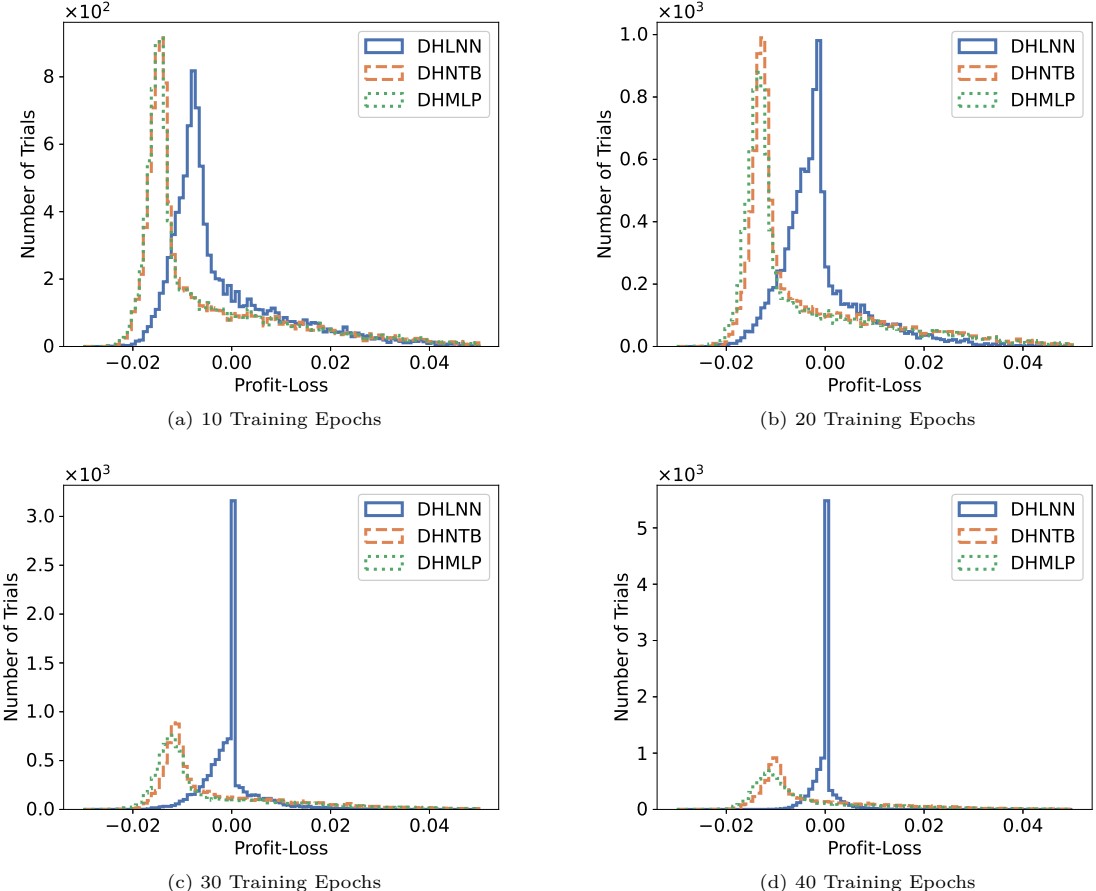

Figure 13: Comparison of convergence performance across different training epochs $\{10, 20, 30, 40\}$ for deep hedging models applied to a Lookback option with a strike price of 1.2. The analysis considers a transaction cost of $2 \times 10^{-3}$.

Moreover, the mixed second-order partial derivatives $\nabla^2_{\boldsymbol{b}^{(l)}, \boldsymbol{W}^{(l)}} f_k$ and $\nabla^2_{\boldsymbol{W}^{(l)}, \boldsymbol{b}^{(l)}} f_k$ are zero due to the structure of the model function and the independence of the weight matrices $\boldsymbol{W}^{(l)}$ and the bias vectors $\boldsymbol{b}^{(l)}$. The weight matrix $\boldsymbol{W}^{(l)}$ and the bias vector $\boldsymbol{b}^{(l)}$ at the same layer $l$ do not directly interact in the second-order sense. Therefore, their partial derivatives are computed independently, resulting in zero mixed second-order derivatives

$$\frac{\partial}{\partial \boldsymbol{b}^{(l)}} \left( \frac{\partial f_k}{\partial \boldsymbol{W}^{(l)}} \right) = 0 \quad \text{and} \quad \frac{\partial}{\partial \boldsymbol{W}^{(l)}} \left( \frac{\partial f_k}{\partial \boldsymbol{b}^{(l)}} \right) = 0. \tag{92}$$

The overall Hessian norm is bounded by considering the maximum bounds of the components

$$\left\| \nabla^2 f_k(\boldsymbol{x}, \boldsymbol{w}, \boldsymbol{a}_k) \right\| \leq \max \left( A(M_2 H^2 + M_1)^L, A M_2^L \right). \tag{93}$$

# E  Performance Comparison with Additional Simulations

## E.1  Convergence Analysis for Lookback Options with Simulated Market Data

Fig. 13 provides a detailed comparison of the convergence performance of DHLNN, DHNTB, and DHMLP for Lookback options with a strike price of 1.2. This analysis evaluates the evolution of PNL distributions across training epochs $\{10, 20, 30, 40\}$ under a transaction cost of $2 \times 10^{-3}$. The purpose of this experiment is to validate the robustness of the proposed DHLNN framework under path-dependent conditions typical of Lookback options.

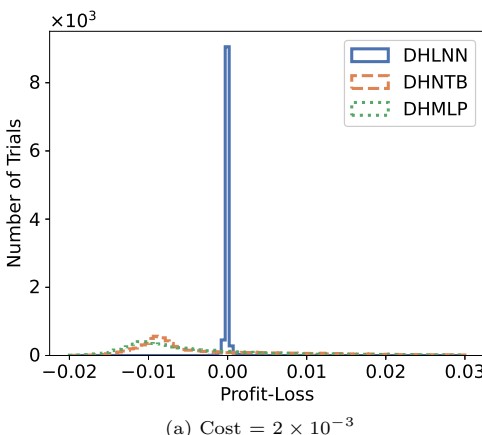 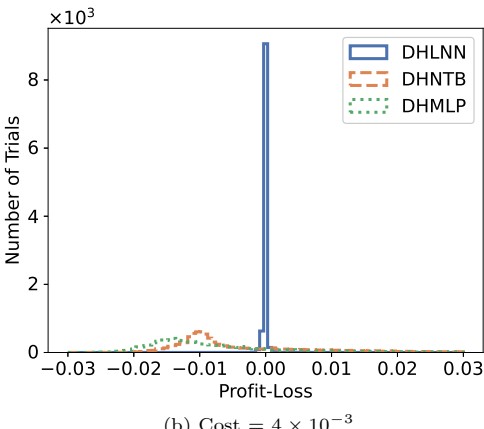

(a) Cost $= 2 \times 10^{-3}$
(b) Cost $= 4 \times 10^{-3}$

Figure 14: Hedging Performance comparison over different transaction costs $\{2 \times 10^{-3}, 4 \times 10^{-3}\}$ of underlying asset for a Lookback option with a strike price of 1.2 and 50 training epochs for the distribution of Hedging PNL where volatility is set as 0.1.

At 10 training epochs, DHLNN already demonstrates a narrower and more concentrated PNL distribution compared to DHNTB and DHMLP. This early-stage performance highlights DHLNN's ability to rapidly adapt to the path dependencies inherent in Lookback options. Both DHNTB and DHMLP, while functional, exhibit broader PNL distributions with significant tail risks, underscoring their slower convergence and susceptibility to noise.

By the 20th epoch, the advantages of DHLNN become more pronounced. The PNL distribution is further refined, with a sharper peak around zero, indicating improved convergence and risk mitigation. DHNTB and DHMLP, though showing some improvement, continue to lag in achieving the same level of stability and precision as DHLNN. At 30 and 40 epochs, DHLNN achieves near-optimal convergence, with its PNL distribution sharply centered around zero and minimal variance. This performance demonstrates DHLNN's ability to effectively neutralize risk and adapt to complex payoff structures, such as those of Lookback options. In contrast, DHNTB and DHMLP continue to struggle with broader distributions, reflecting residual risks and less efficient optimization.

## E.2 Robustness to Market Frictions on Lookback Options with Simulated Market Data

Fig. 14 evaluates the robustness of DHLNN, DHNTB, and DHMLP under varying transaction costs $\{2 \times 10^{-3}, 4 \times 10^{-3}\}$ for Lookback options with a strike price of 1.2. This experiment highlights the models' ability to adapt to market frictions and manage risk effectively when transaction costs increase. The goal of this analysis is to assess the sensitivity of each model's PNL distribution to varying cost levels and to evaluate their effectiveness in minimizing financial risk under simulated market conditions.

For a transaction cost of $2 \times 10^{-3}$, DHLNN exhibits a sharply concentrated PNL distribution centered around zero, indicating its strong ability to neutralize risk while efficiently managing costs. In contrast, DHNTB and DHMLP display broader PNL distributions with significant tails, reflecting higher exposure to over-hedging and under-hedging risks. The results underscore the inefficiencies of the baseline models in handling transaction costs, particularly when market frictions are relatively low. As the transaction cost increases to $4 \times 10^{-3}$, the performance gap between DHLNN and the baseline models becomes even more apparent. While DHLNN maintains a narrow and stable PNL distribution with minimal variance, DHNTB and DHMLP show further deterioration in their performance, characterized by wider distributions and increased tail risks. This result demonstrates DHLNN's robustness to higher transaction costs, showcasing its ability to achieve consistent risk-neutral hedging strategies even in challenging market conditions.

Fig. 15 provides a comprehensive evaluation of the robustness of DHLNN, DHNTB, and DHMLP under varying transaction costs $\{2 \times 10^{-3}, 4 \times 10^{-3}, 6 \times 10^{-3}, 8 \times 10^{-3}\}$ for Lookback options with a strike price of 1.2. The performance is assessed using two Entropic Loss as shown in Fig. 15(a), which captures overall risk,

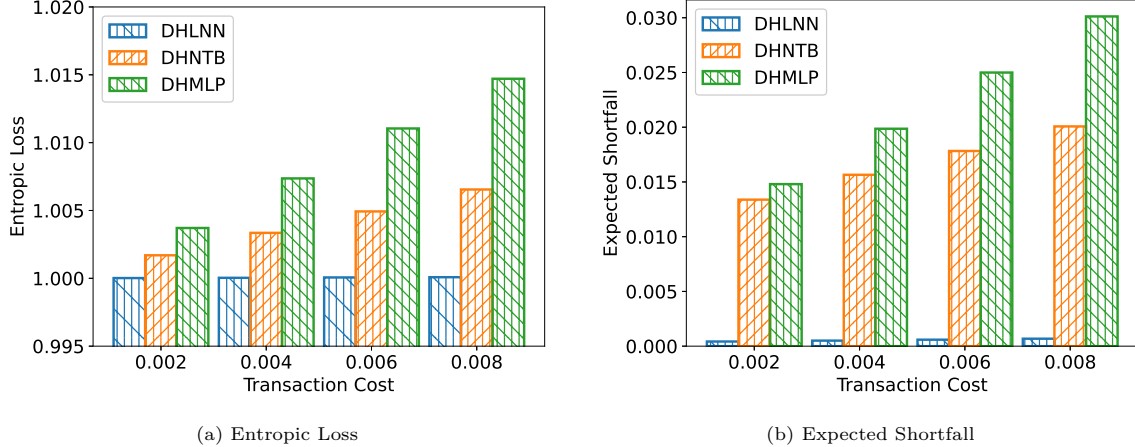

(a) Entropic Loss            (b) Expected Shortfall

Figure 15: Hedging Performance comparison over different transaction costs $\{2 \times 10^{-3}, 4 \times 10^{-3}, 6 \times 10^{-3}, 8 \times 10^{-3}\}$ of the underlying asset for a Lookback option with a strike price of 1.2 and 50 training epochs, evaluated using Entropic Loss and Expected Shortfall of Hedging PNL where volatility is set as 0.1.

and Expected Shortfall as shown in Fig. 15(b), which evaluates tail risk. The purpose of this analysis is to determine the resilience of each model under market frictions introduced by increasing transaction costs and to validate the practical applicability of the proposed DHLNN framework.

In terms of Entropic Loss, DHLNN consistently achieves the lowest values across all transaction cost levels, highlighting its ability to minimize overall risk effectively. Even as the transaction costs increase, DHLNN exhibits only a marginal rise in Entropic Loss, showcasing its robustness and efficient adaptation to higher frictions. In contrast, DHNTB and DHMLP demonstrate significantly higher Entropic Loss values, with a pronounced increase as transaction costs rise. These results suggest that the baseline models are less capable of handling market frictions, leading to suboptimal risk management strategies.

For Expected Shortfall, DHLNN again outperforms the baseline methods, achieving the lowest values across all transaction cost scenarios. The steady and minimal increase in Expected Shortfall for DHLNN indicates its ability to maintain effective risk mitigation in adverse conditions, minimizing extreme losses. DHNTB and DHMLP, on the other hand, show a steep rise in Expected Shortfall, indicating higher vulnerability to tail risks as transaction costs increase.

