# OpenReview forum: "Robust and Efficient Deep Hedging via Linearized Objective Neural Network"
_TMLR — Rejected by TMLR_

### Review · Reviewer_qSY5 · 2024-10-09

**Summary Of Contributions:**

The paper focuses on deep hedging, which consists of applying deep neural networks to financial derivatives for risk management. The authors tackle the computationally expensive training by applying a linearized network architecture combined with the Black-Scholes Delta framework, which assumes derivatives are replicated by underlying instruments in an idealized market. The authors claim to enhance the efficiency and interpretability of hedging strategies via this linearized approach and address action-dependence challenges with faster convergence. They also perform experiments where their method shows advanced performance compared to other state-of-the-art deep hedging models.

**Audience:**

Yes

**Broader Impact Concerns:**

Does not apply.

**Claims And Evidence:**

No

**Requested Changes:**

Although the authors address an interesting application in financial risk management with deep hedging, key issues related to the clarity of the background, the justification of experimental design choices, details for reproducibility, and the presentation of results raise significant concerns regarding their findings' overall reliability and interpretability. based on this, the following changes could be beneficial for the presentation of their work:
1. **References and Explanations for Financial Concepts:** based on [W1, W2], the reading quality of the paper could be improved by making clearer the several introduced concepts in terms of financial risk management. For instance, in [1], authors clarify the relevant definitions with numbering and titles. It would be helpful to more clearly differentiate between the background information and the novel contributions in the methods section. Supporting comments and introducing concepts with references are crucial for the understanding of non-experts in finance.

2. **Experimental Design and Reproducibility:** based on [W3, W4, W6], the results section could benefit from precise explanations for hyperparameter choices in the experimental design and the presented results.

3. **Results Presentation, Formatting, and Appendix:** based on comments [W5, W7], the quality of visualization could be improved by different design choices/grouping of figures / better explanation of key differences (e.g., in captions). Similarly, the appendix could include additional information/more content from the main paper (e.g., some results).

**Strengths And Weaknesses:**

The main strengths and weaknesses of the paper are summarized as follows:

Strengths:
- **[S1]:** The authors thoroughly analyze the optimal hedging problem in a whole section of the paper, followed by their proposed method. Finally, they provide an analysis section for the approximation quality of the hedging model, with regards to the magnitude and changing rate of the gradient, as well as the bound of the hessian.
- **[S2]:** The proposed model is designed to be robust and explainable via the introduced linearized neural network while being adjustable to various utilities and derivatives. Additionally, the authors provide the details of the approximation by linearization.
- **[S3]:** Experimental results show their method’s effectiveness under several market conditions in terms of convergence speed and performance metrics against relevant deep hedging methods.

Weaknesses:
- **[W1]:** Significant concepts in terms of finance are not adequately presented and supported by references. Improving this aspect is significant for people from the machine learning community who are interested in learning about this application. Relevant examples are the first paragraph of the “Related Work” section (page 2) and the second paragraph of the “Introduction” section (page 2).
- **[W2]:** Following the above, it would be very helpful for people non-experts in finance that significant terms introduced are also accompanied by references, e.g., WAP definition and statements on it (3.1), risk measures, and statements about them (3.3), log moneyless and volatility (4.1), anchor hedge strategy and why use this upon others (4.4). Similarly, several statements about strategies and choices accompanied by explanations on the financial market’s side could be further supported by references. A few notable examples are provided: “Consequently, hedgers expose themselves […] premium paid”, “Maturity refers to […] at this end date”, “All trading operations are self-financed […] the hedging strategy”, “The key innovation lies in transforming […] in transaction costs”.
- **[W3]:** In the first part of the experiments, the authors use synthetic data following a geometric Brownian motion. Additional information about the hyperparameters of the stochastic needs to be determined to enable reproducibility of the results. Are the train and test distributions significantly similar or not? Could the experimental analysis be more thorough if additional experiments, including out-of-distribution testing for the synthetic data generation, be explored?
- **[W4]:** The $10,000$ Monte Carlo trajectories seem not to match relevant studies [1] and, similarly, the number of epochs used for training. Additionally, the number of iterations for Monte Carlo is not mentioned. It is important for the authors to justify their choices and clarify how this affects the market dynamics and the performance of the model.
- **[W5]:** In the results section, the authors present a total of $13$ figures in the main paper. Unfortunately, all of them have the same colors and style in terms of the plots, with minor differences in the y-axis or x-axis variables/measures, which makes the general message and differentiation between results extremely hard. Figures should include a description of the metric captured and the ideal values (e.g., lower is better). Finally, figures capturing PNL distributions are hard to read since several segments overlap.
- **[W6]:** In the presented results, significant choices are not adequately justified, making it very difficult for non-exerts in finance to assess the significance of performance and the effect of the choice. For instance, transaction costs are selected in $\{2 \times 10^{-3}, 4 \times 10^{-3}, 6 \times 10^{-3}, 8 \times 10^{-3}\}$, a strike price of $1.2$ or $1.0$, volatility of $0.1$ or different volatility scenarios of $0.1, 0.2, 0.3, 0.4$.
- **[W7]:** More Issues: References in the text are not in parenthesis (messing with expressions). Additionally, in the Appendix, there should be some text accompanying the additional figures provided to make their message clearer.

[1] Imaki, S., Imajo, K., Ito, K., Minami, K., & Nakagawa, K. (2021). No-transaction band network: A neural network architecture for efficient deep hedging. arXiv preprint arXiv:2103.01775.

---

> ### Author Response · Authors · 2025-01-16
> **Presentation and Support of Financial Concepts**
>
> We appreciate the reviewer's observation regarding the need to present financial concepts more comprehensively and with stronger references. In the revised manuscript, we have thoroughly rewritten the Introduction and Related Work sections to present key financial ideas while supporting them with appropriate references to foundational works. This revision aims to make the manuscript easier to follow and more interesting for readers from the machine learning community who may not be deeply familiar with financial applications.
>
> In the revised Introduction, we clearly illustrate the limitations of traditional financial methods, such as the Black-Scholes framework, in addressing challenges like path dependency, transaction costs, and nonlinear relationships. We explain how deep learning, and specifically deep hedging, offers a flexible and data-driven approach to manage these complexities, referencing key foundational works. Additionally, we detail persistent challenges in implementing deep hedging frameworks, such as computational inefficiency, sensitivity to noisy market data, and optimization complexity, while motivating the need for our proposed innovation.
>
> The Related Work section has also been significantly revised to establish a stronger connection between financial optimization challenges and the methods proposed in recent machine learning advancements. We discuss the limitations of existing heuristic-driven and hybrid methods, highlighting the need for interpretability and scalability in financial applications.
>
> To improve clarity and relevance for the machine learning audience, we have restructured the presentation to focus the main text on high-level concepts and their relevance to machine learning applications, while moving detailed financial explanations to the Appendix. This approach ensures that readers can quickly grasp the core ideas and innovations, with additional depth available for those seeking a more detailed understanding of financial methods. The revised manuscript emphasizes that our proposed framework, Deep Hedging with Linearized Objective Neural Network (DHLNN), introduces a transformative approach to training deep learning models in financial applications. By enhancing computational efficiency, training stability, and robustness, DHLNN addresses the challenges faced by traditional methods. These revisions underscore the generalizability of our innovations, positioning DHLNN as a significant advancement not only in financial risk management but also in broader machine learning applications.

---

> ### Author Response · Authors · 2025-01-16
> **Enhanced Explanations and References for Financial Terms and Strategies**
>
> Thank you for pointing out areas where additional clarification and detail would benefit readers who may not have a background in finance. In response to the concerns raised, we have rewritten Section 3 entirely to provide a clearer and more structured explanation of the hedging procedures and their underlying objectives. The revised section now offers a comprehensive discussion on dynamic hedging strategies, focusing on how they operate along the price trajectory of underlying assets during the option's lifetime.
>
> To facilitate understanding for readers from the machine learning community, we include a simple yet illustrative example that explains the hedging process. This example demonstrates the application of dynamic hedging to mitigate the liability that arises at the maturity of an option, which represents the financial risk borne by the option writer. The objective of the hedging strategy is explicitly framed as determining the optimal position in the underlying assets to neutralize this risk effectively.
>
> Recognizing the need for easy understanding, we have also moved detailed definitions of key financial concepts, such as Weighted Average Price, convex risk measures, and the indifference pricing framework, into Appendix A, B, and C. This adjustment ensures that readers unfamiliar with financial terminology can grasp the high-level objectives and innovations of our work without being overwhelmed by technical jargon.
>
> Regarding the anchor hedge strategy introduced in Section 4, we emphasize that this component serves as a practical enhancement to align the outputs of trained deep learning models with established financial principles. Integrating the Black-Scholes Delta as an anchor not only improves the interpretability of the model but also demonstrates how traditional financial models can complement modern deep learning methods. It is important to note that while the Black-Scholes Delta anchor represents one specific application, the concept of anchoring can be generalized and integrated with various financial models. Our primary contribution lies in the development of a novel linearized training procedure that fundamentally improves the efficiency, robustness, and explainability of deep learning-based hedging strategies. The integration of the Black-Scholes Delta is an illustrative case study that highlights how traditional financial theory can be leveraged to enhance the practicality and reliability of machine learning models in this domain.

---

> ### Author Response · Authors · 2025-01-16
> **Clarification and Expansion of Synthetic Data Settings and Experiments**
>
> We appreciate the reviewer's insightful comments. In the revised manuscript, we have thoroughly updated the description of the synthetic data generation process to ensure transparency and reproducibility. Some details are as follows:
>
> In the first part of our experiments, we utilize synthetic data generated through a geometric Brownian motion model, consistent with the notations and formulations in the paper. The stochastic price dynamics are given by
> \begin{equation}
> dP_{t_i} = \mu P_{t_i} dt + \sigma P_{t_i} dW_{t_i},
> \end{equation}
> where $P_{t_i}$ represents the price of the underlying asset at time $t_i$, $\mu$ is the drift term, $\sigma$ is the volatility, and $dW_{t_i}$ is an increment of a Wiener process. We set $\mu = 0$, ensuring a martingale process. The trajectories are simulated over a maturity $T = 30/365$ years, i.e., 30 days, using a daily time step $dt = 1/365$. For each path, the stochastic increments $dW_{t_i}$ are modeled as standard normal random variables scaled by $\sqrt{dt}$. A correction term $\frac{\sigma^2 t_i}{2}$ is subtracted to adjust for the expected drift, ensuring the price evolution remains consistent with the risk-neutral dynamics
> \begin{equation}
> P_{t_i} = P_0 \exp\left(B_{t_i} - \frac{\sigma^2 t_i}{2}\right),
> \end{equation}
> where $B_{t_i}$ is the cumulative Brownian motion. For all experiments, the volatility $\sigma$ is set to $0.1$ during training, while the testing phase incorporates higher volatilities $\sigma \in \{0.2, 0.3, 0.4\}$ to evaluate robustness. The initial price $P_0$ is normalized to $1$, and $10,000$ independent paths are generated for training and testing at each epoch. This experimental setup is designed to rigorously evaluate the performance of the proposed DHLNN framework under both in-distribution and out-of-distribution conditions. By training the model on data generated with a fixed volatility of 0.1 and testing it on scenarios with significantly higher volatilities 0.2, 0.3, and 0.4, we gain valuable insights into the framework's adaptability and resilience across diverse market conditions. This out-of-distribution evaluation highlights the robustness of DHLNN in adapting to dynamic and unforeseen changes in market volatility, demonstrating its capacity to generalize effectively beyond the specific conditions seen during training.
>
> To ensure a thorough analysis, we have explicitly incorporated out-of-distribution testing for the synthetic data generation in Section 5.3.3 of the revised manuscript. These experiments provide a detailed examination of how the model performs when confronted with market conditions that deviate from the assumptions underlying the training data. By extending the analysis to include out-of-distribution scenarios, we emphasize the practical applicability of DHLNN in real-world financial environments, where market conditions often exhibit volatility patterns and dynamics that are challenging to predict. This addition strengthens the experimental evaluation and addresses the  concern regarding the need for a more comprehensive exploration of the framework's robustness.

---

> ### Author Response · Authors · 2025-01-16
> **Justification of Monte Carlo and Training Configurations**
>
> To address the reviewer's comment regarding the Monte Carlo trajectories and the number of epochs used for training, as well as to justify our method choices in light of the referenced study [1], we provide the following clarifications:
>
> Our experimental setup employs $10,000$ Monte Carlo trajectories for the price dynamics of the underlying asset, generated using a geometric Brownian motion model. This choice reflects a balance between computational efficiency and the need to capture sufficient diversity in the simulated data for training and validating the hedging model. While the referenced study [1] uses $50,000$ trajectories, it is important to note that their focus lies on a different architecture tailored to address specific action-dependence issues. In our case, $10,000$ trajectories are sufficient to ensure robust and efficient performance evaluation while maintaining manageable computational costs, aligning with the objectives of our study.
>
> The training process in our experiments spans $50$ epochs for most scenarios. This choice ensures convergence and stability of the DHLNN model while avoiding overfitting. During training, performance metrics such as PNL, Entropic Loss, and Expected Shortfall are monitored across epochs to validate the stability and effectiveness of the training process. Our experiments confirm that $50$ epochs provide sufficient iterations for the model to converge effectively, given the well-designed architecture and optimization techniques, including our linearized training procedure. In comparison to [1], where a larger number of epochs may have been employed, our design prioritizes computational efficiency while ensuring robust results.
>
> We also emphasize the importance of evaluating the model's robustness under out-of-distribution conditions. In our experiments, the trained model is tested across varying volatility regimes $\sigma = 0.2, 0.3, 0.4$, enabling a comprehensive analysis of its adaptability and robustness. These out-of-distribution evaluations provide critical insights into how the model generalizes to unseen market conditions, reinforcing its applicability to real-world financial scenarios.
>
> While our choices regarding the number of trajectories and training epochs differ from those in [1], they are well-justified given the unique objectives and constraints of our study on evaluating the efficiency and robustness. Our results demonstrate that the proposed approach achieves a strong balance between computational efficiency and model performance.

---

> ### Author Response · Authors · 2025-01-16
> **Improved Figure Presentation and Differentiation**
>
> We sincerely thank the reviewer for the constructive feedback on the presentation of figures in the results section. To address the concerns regarding the clarity, differentiation, and interpretability of the figures, we have made significant improvements in the revised manuscript. These changes aim to enhance the visual clarity of each figure, provide explicit descriptions of the metrics, and ensure that the results are easily interpretable, particularly for readers from diverse backgrounds.
>
> In the revised figures, we have included detailed captions that clearly explain the metrics being evaluated, their significance, and the ideal values for comparison. For example, in figures displaying Profit and Loss (PNL) distributions, the captions now explicitly state that “a narrower and more centered distribution around zero indicates superior hedging performance.” This ensures that readers can immediately understand the evaluation criteria and the intended takeaway from the figures.
>
> To differentiate the figures visually, we have introduced distinct hatching patterns, such as stripes and grids, along with varying marker styles for different models in the bar plots. This approach maintains consistent colors for specific methods across all figures while improving visual distinction. For figures with overlapping PNL distributions, we have employed transparent overlays and fine-grained histograms to reduce visual clutter and improve interpretability. These changes make it easier for readers to distinguish between methods and understand their relative performance.
>
> Additionally, the axis labels and legends in each figure now explicitly describe the metric captured and indicate whether lower or higher values are better. For instance, figures illustrating Entropic Loss and Expected Shortfall metrics include notes in the captions emphasizing that “lower values indicate better performance.” In the case of PNL distribution figures, annotations explain that distributions centered around zero with minimal variance signify effective hedging with reduced risk exposure.
>
> To enhance the logical flow of the results section, the figures are now organized into subsections that align with specific experimental objectives, such as convergence analysis, robustness testing and out-of-distribution evaluations. Each subsection introduces the figures with a brief description of the experimental purpose, the metrics being assessed, and the key insights derived. This restructuring ensures that readers can easily follow the narrative and understand how each figure supports the experimental objectives.

---

> ### Author Response · Authors · 2025-01-16
> **Justification of Experimental Parameter Choices**
>
> We greatly appreciate the reviewer's feedback regarding the need to justify the significant choices made in our experiments, particularly in terms of transaction costs, strike prices, and volatility scenarios. These parameters were carefully chosen to evaluate the performance of the proposed DHLNN framework across a range of representative market conditions, while maintaining computational feasibility.
>
> The selected transaction cost rates, i.e., $2 \times 10^{-3}$, $4 \times 10^{-3}$, $6 \times 10^{-3}$, and $8 \times 10^{-3}$, were chosen to represent proportional costs associated with trading the underlying asset. These values are consistent with typical market frictions observed in real-world financial markets, including bid-ask spreads and execution costs, and span a range from low to moderately high transaction costs. This range allows for a comprehensive evaluation of the robustness and efficiency of hedging strategies under varying levels of market frictions. By analyzing the performance across different cost scenarios, we aim to highlight the trade-offs between risk minimization and cost efficiency, which are central considerations in financial optimization.
>
> The strike prices of $1.0$ and $1.2$ were selected to represent different market scenarios, including derivatives that are at-the-money (ATM), or out-of-the-money (OTM). A strike price of $1.0$ reflects an ATM scenario when the initial price of the underlying asset is close to the strike price, capturing scenarios with balanced risk exposure. A strike price of $1.2$ represents an OTM option at the start, creating a scenario with higher risk exposure for the writer. This diversity ensures that the adaptability and robustness of the DHLNN framework are rigorously tested across a range of market conditions.
>
> Volatility was set at levels ranging from $0.1$ to $0.4$ to represent diverse market dynamics. Volatility directly impacts derivative pricing and the complexity of hedging, with $0.1$ representing relatively stable markets and $0.4$ corresponding to highly volatile conditions. The training data was generated with a fixed volatility of $0.1$, reflecting calm market conditions, while testing was conducted across higher volatility scenarios $0.2$, $0.3$, and $0.4$ to evaluate the framework's robustness under out-of-distribution conditions. This out-of-distribution analysis is particularly significant for assessing the generalizability of the proposed model in dynamic and unpredictable market environments.
>
> To address the reviewer's concerns and support reproducibility, we have revised the manuscript to include a detailed explanation of the rationale behind these experimental choices in Section 5.

---

> ### Author Response · Authors · 2025-01-16
> **Formatting and Clarity Issues in References and Appendix**
>
> We appreciate the reviewer's observation regarding the formatting of references and the clarity of the additional figures in the Appendix. These issues have been carefully addressed in the revised manuscript to ensure clarity and professionalism in the presentation. Regarding the references in the text, we have corrected all instances where references were not properly formatted in parentheses. For the figures provided in the Appendix, we have added accompanying descriptive text for each figure to clarify their purpose and significance.

---

> ### Author Response · Authors · 2025-01-16
> **Summary to the Requested Changes**
>
> To enhance clarity for non-experts, we revised the Introduction and Related Work sections, clearly distinguishing between background concepts and novel contributions while providing accessible explanations of financial terms, supported by references. Key financial definitions and examples, such as hedging strategies and risk measures, are included in Section 3, with more technical details moved to the Appendix for accessibility. In the Experimental section, we detailed the settings for synthetic data generation, model architecture, and hyperparameters, justifying choices such as transaction costs, strike prices, and volatility scenarios to ensure reproducibility and relevance. Out-of-distribution testing was added to evaluate robustness under diverse market conditions. Figures were redesigned for clarity, with improved visual styles and captions that explicitly describe metrics and ideal outcomes, while supplementary figures in the Appendix now include explanatory text.

---

### Review · Reviewer_bCNU · 2024-11-14

**Summary Of Contributions:**

This paper seems to propose a novel architecture to incorporate Black Schole Delta with neural network design for deep hedging task.
Their experimental results are quite strong compared to the earlier baselines.

**Audience:**

Yes

**Broader Impact Concerns:**

I do not have any concerns.

**Claims And Evidence:**

No

**Requested Changes:**

See above  weaknesses

**Strengths And Weaknesses:**

Summary:

This paper proposes to use linear neural network to solve deep hedging problem.

Specifically, it try to design a method to incorporate Black-Scholes’ Delta to mitigate the challenge of action-dependence and support high-volume orders and rapid injection of liquidity into derivative markets.


I begin with three questions:

What is the current state of deep hedging field?
Why linear neural network instead of other neural architectures?
Why the authors claim it is robust?

After reading this paper, I found the current presentation prevent me understand the logic of their method.

Maybe the authors should consider reorganize their paper into
- What is the basic concept of deep hedging
- what is the actual design of their method, in plain words without convoluted equations
	- Show the differences of their approach with baselines
- Reorganize their analysis into why their method is robust




# Other comments during the reading process of this paper:

## Introduction

It seems, in the introduction, relate the deep hedging is equivalent to finding an optimized state of given neural network. For example NTB.

In addition, why citing AlexNet to show task specific design of neural architecture is beneficial? I am not sure if the authors try to say “what the people in deep hedging domain should also design task specific model?” Is this like a common trend already, as cited Buehler et al 2019 work and the Imaki et al. 2021 already designed NTB for this?

## Related work

Citations are mixed. The authors should learn \citet and \citep to properly address the citation issues.


You should probably mention the full name of DHLNN in page 3, as it is only introduced once in the abstract.


## Hedging optimization

You should consider adding a figure to better illustrate the hedging transactions. After all, TMLR is a machine learning journal, many readers lacks sufficient finance knowledge to comprehend the materials.

It is well structured, but maybe the authors can consider moving some of the derivation steps and keeps the conclusion of the final optimization goal (Eq.23). Instead just laying out those derivations, maybe it is better to explain the Eq 23 in words.

## Deep hedging formulation

I am confused, the 4.2 seems to be an introduction of linear neural networks alone, but how does this relates to the deep hedging as in Section 3? In short, in Eq. 24, it uses notation of h, W, x, b. But, how does this relates to section 3?

Then it goes to 4.3.1 talking about training dynamics, where again it seems to be pure neural networks optimization, without relation to hedging.

Then in section 4.3.2 Linearized hedging model, it says “Hessian bounded and gradient do not change rapidly so critical for the design”. What design?

Then Eq. 37 says ”according to the analysis in Section 5, we define beta = …”
Eq. 37 involves A M H. What are these notations for?

Up to now, I am honestly lost. In 4.4, it seems to relate the logic of Black-Scholes Delta, and all notation is about those defined in section 3 hedging optimization.

I think probably this is a domain gap between the deep hedging domain. But in terms of a machine learning aspect, these section is not readable. Probably the authors should consider delineate their formulation in a framework figure, connecting all the dots into a complete logic flow for better comprehension.
In addition, it should also consider moving many derivation steps into the appendix, keeping the most useful conclusion in the text.

---

> ### Author Response · Authors · 2025-01-16
> **Current State of Deep Hedging Research**
>
> The field of deep hedging has seen significant advancements in recent years, emerging as a transformative approach to financial risk management. Traditional hedging frameworks, such as the Black-Scholes model, are limited by their reliance on closed-form solutions and simplifying assumptions, which restrict their applicability in complex market scenarios. In contrast, deep hedging leverages the power of machine learning, particularly deep neural networks, to model and optimize hedging strategies dynamically. This approach enables the handling of diverse market frictions, noisy data, and intricate relationships between financial variables.
>
> Key developments in the deep hedging field include the integration of advanced neural architectures to capture dynamic market behaviors and optimize trading decisions. Furthermore, deep hedging frameworks increasingly incorporate financial domain knowledge, such as model-free approaches or risk-sensitive objectives, to enhance interpretability and alignment with established financial theory. Practical applications have expanded beyond equity derivatives to include commodities, currencies, and credit products, demonstrating the versatility of deep hedging in managing financial risks.
>
> Despite its promise, the field faces challenges, including high computational costs, sensitivity to data quality, and the need for robust and explainable models that can meet regulatory requirements. We have focused on improving training efficiency, stability, and robustness through innovations such as linearized optimization techniques, model regularization, and hybrid frameworks combining machine learning with traditional financial models. These advancements position deep hedging as a key area for future exploration, with the potential to address emerging challenges in financial risk management and broaden its impact across diverse markets.

---

> ### Author Response · Authors · 2025-01-16
> **Justification for Using Linearized Objective Neural Network**
>
> Our selection of a linearized neural network objective is driven by its unique capability to tackle the challenges inherent in financial risk management, while simultaneously introducing a transformative innovation in the training process. Central to our work is a novel linearized training procedure, which fundamentally redefines the optimization dynamics of deep learning models for financial hedging tasks. This is achieved through a nested training framework that incorporates periodic fixed-gradient updates, enabling the objective function to be linearized during inner iterations.
>
> The linearized training procedure stands out as a pivotal advancement in our framework. By decoupling gradient computation from parameter updates, this approach stabilizes the optimization landscape, allowing multiple optimization steps to be performed on a linearized version of the objective function. This significantly reduces computational complexity while accelerating convergence, ensuring that the training process is both efficient and robust. Unlike conventional deep learning methodologies, which often struggle with the challenges of non-convex optimization landscapes and sensitivity to noisy data, our approach stabilizes training dynamics and delivers consistent performance across a wide range of market scenarios.
>
> The use of a linearized neural network objective further enhances the model's ability to manage noisy data and navigate volatile market conditions, both of which are common in real-world financial environments. By creating a stable optimization framework, the linearized training procedure mitigates overfitting and improves the model's generalizability, making it particularly effective in handling out-of-distribution market scenarios.
>
> Our key innovation lies not only in adopting a linearized neural network objective but also in the development of the linearized training procedure. This nested optimization framework seamlessly integrates computational efficiency, rapid convergence, and enhanced interpretability, establishing a new standard for deep learning models in financial risk management. By addressing the complexities of financial hedging tasks, our approach offers a robust, scalable, and explainable solution tailored to the dynamic and uncertain nature of financial markets.

---

> ### Author Response · Authors · 2025-01-16
> **Robustness of the Proposed Framework**
>
> The robustness of our proposed framework stems from several key features that directly address the inherent challenges of financial risk management, such as noisy data, volatile market conditions, and complex optimization landscapes. These features include the innovative linearized training procedure, the nested optimization framework, and the integration of established financial principles, such as the Black-Scholes Delta, to enhance stability and interpretability.
>
> Our claim of robustness is supported by the novel linearized training procedure, which stabilizes the optimization process through periodic fixed-gradient updates. This approach linearizes the objective function during inner iterations, creating a stable and simplified optimization landscape. By decoupling gradient computation from parameter updates, the model reduces sensitivity to noisy data and ensures consistent convergence even under dynamic and volatile market conditions. This stability enhances the model's generalizability and makes it well-suited for both in-distribution and out-of-distribution scenarios, as demonstrated in our experiments.
>
> Furthermore, the nested optimization framework allows the model to perform multiple updates on the linearized objective before recalculating gradients, reducing computational complexity while maintaining accuracy. This design ensures that the model remains robust against overfitting, adapts effectively to changing market environments, and consistently delivers reliable hedging strategies.
>
> In addition to its methodological innovations, our framework incorporates the Black-Scholes Delta as an anchor for the hedging strategy. This integration provides a theoretically grounded reference point that guides the model's outputs, improving explainability and ensuring alignment with well-established financial principles. By combining data-driven techniques with financial theory, the model effectively balances flexibility and precision, further contributing to its robustness.
>
> The robustness of the proposed framework is thoroughly validated through extensive experiments on both synthetic and real market data. Our results consistently show that the model achieves narrow, stable, and centered Profit and Loss distributions, even under high transaction costs and volatile market conditions. Moreover, the framework's ability to maintain low entropic loss and expected shortfall across diverse scenarios demonstrates its capacity to handle extreme market events and mitigate tail risks.

---

> ### Author Response · Authors · 2025-01-16
> **Clarification and Reorganization of the Method Presentation**
>
> We sincerely thank the reviewer for the insightful recommendations on reorganizing the manuscript to enhance clarity and accessibility. In response to these suggestions, we have made substantial revisions to the paper, restructuring it to better align with the proposed framework and ensure the logical flow of ideas.
>
> To address the recommendation to provide a clear explanation of the basic concept of deep hedging, we have expanded the first 3 sections. These revised sections now offer a detailed overview of the principles of deep hedging, highlighting how it extends traditional financial risk management methods. We ensure that readers, including those new to the topic, can understand the motivations behind our work.
>
> The Methodology section has been revised to include a plain-language explanation of our proposed framework, Deep Hedging with Linearized Objective Neural Network (DHLNN). Before introducing mathematical details, we provide an intuitive overview of the method's key innovations, such as the linearized training procedure, nested optimization framework, and Black-Scholes Delta anchoring. By describing these components in accessible terms, we make the framework more comprehensible to readers who may not have a background in finance.
>
> To emphasize the robustness of our method, we have reorganized the experimental results section. This section now highlights how DHLNN performs under various conditions. The analysis attributes the model's robustness to the linearized training procedure and nested optimization framework, providing clear evidence of its advantages in dynamic and challenging market scenarios.
>
> We have also streamlined the main text by moving detailed explanations of financial concepts to the Appendix. This adjustment ensures that the primary narrative remains focused on the key ideas and innovations, making the manuscript more accessible while providing sufficient context for readers seeking additional detail.

---

> ### Author Response · Authors · 2025-01-16
> **Clarifying the Relationship Between Deep Hedging and Neural Network Optimization**
>
> We appreciate the reviewer’s observation regarding the connection between deep hedging and neural network optimization. Deep hedging, as a broad concept, encompasses a variety of approaches aimed at optimizing hedging strategies using machine learning techniques. The No-Transaction Band (NTB) is indeed one such approach within the deep hedging framework. In the NTB methodology, trading activity is constrained within predefined thresholds to reduce transaction costs, making it an effective solution for specific scenarios. However, the NTB framework represents just one implementation of deep hedging and does not fully encapsulate the concept as a whole.
>
> In the revised introduction, we clarify that deep hedging is not merely about optimizing a given neural network, but about designing and training models to navigate the complexities of financial markets dynamically. This involves integrating neural networks with principles of financial optimization to address challenges like transaction costs, market frictions, path dependencies, and volatile market conditions.
>
> Our proposed DHLNN builds upon the core ideas of deep hedging but introduces significant innovations to overcome key limitations of existing approaches. Specifically, DHLNN employs a nested optimization framework with linearized training dynamics to improve computational efficiency, stabilize training, and enhance robustness. This design enables it to adapt dynamically to changing market conditions while reducing sensitivity to noisy data.

---

> ### Author Response · Authors · 2025-01-16
> **Justification for Citing AlexNet and Task-Specific Neural Architectures**
>
> We appreciate the reviewer's question about the relevance of citing AlexNet to support the idea of task-specific neural architecture design. The inclusion of a reference to AlexNet in the original draft was meant to draw an analogy from the broader machine learning field, highlighting the benefits of task-specific neural architectures. However, we acknowledge the need to clarify the connection to deep hedging and address the reviewer's concerns about the relevance and commonality of task-specific model design in this domain.
>
> In the field of deep hedging, the design of task-specific models is indeed an emerging trend. These studies demonstrate the importance of tailoring architectures and methodologies to the unique challenges of financial markets, such as transaction costs, market frictions, and path dependencies. For example, Imaki et al. introduced the NTB approach, which is specifically designed to reduce transaction costs by implementing no-trading bands, a practical and task-specific solution in the deep hedging domain.
>
> Our work aligns with this trend by introducing the linearized training procedure within the DHLNN framework. This approach directly addresses the challenges of computational inefficiency, training instability, and robustness in financial markets. While our focus is on the training dynamics rather than the architecture itself, this innovation is conceptually similar to task-specific design as it adapts the model's behavior to the unique demands of the financial hedging task.
>
> The analogy to AlexNet was intended to underscore the broader principle that neural network design can benefit significantly from being adapted to the requirements of specific tasks. However, we agree that this example might not resonate as strongly with readers in the deep hedging domain. In our revised manuscript, we have removed the reference to AlexNet and instead focused on the parallels within the deep hedging literature.

---

> ### Author Response · Authors · 2025-01-16
> **Citation Formatting and Full Terminology Clarification**
>
> We appreciate the reviewer's comments on citation formatting and the clarification of terms. The references have been reviewed and corrected, ensuring consistency and proper citation formatting throughout the manuscript.
>
> We have introduced the full name of DHLNN (Deep Hedging with Linearized Objective Neural Network) in our main context to ensure clarity.

---

> ### Author Response · Authors · 2025-01-16
> **Adding a Figure to Illustrate Hedging Transactions**
>
> We appreciate the reviewer's suggestion to include additional materials that better illustrate the hedging transactions for readers with limited finance knowledge. To address this, we have extensively revised Section 3 of the manuscript to provide a comprehensive explanation of the hedging context and objectives. This revision includes a clear and detailed example that demonstrates the mechanics of dynamic hedging in a manner accessible to readers from a machine learning background.
>
> In the revised Section 3, we illustrate the hedging procedure with a detailed example involving a European call option. This example walks through the step-by-step adjustments of the hedging position (hedge delta) over time, guided by the Black-Scholes formula. The figure associated with this example (Fig.1) visually depicts the price trajectory of the underlying asset, the dynamically adjusted hedge position, and the resulting liability and PNL. Additionally, the example explains the practical implications of dynamic hedging, such as reducing liability exposure before maturity. It also emphasizes the importance of intermediate hedging actions along the price trajectory, showcasing how continuous adjustments manage risk effectively. By omitting the final hedging adjustment at maturity in the illustration, we further highlight the cumulative impact of dynamic hedging throughout the trajectory.

---

> ### Author Response · Authors · 2025-01-16
> **Streamlining Derivations and Emphasizing Key Results**
>
> We appreciate the reviewer's suggestion to improve the presentation of derivations and key results. In the revised Section 3, we have intentionally adopted a high-level explanation to make the hedging objective and optimization goal accessible to a broader audience, especially readers with a machine learning background who may not have prior expertise in financial mathematics.
>
> The section now emphasizes the fundamental concepts and practical implications of the hedging objective in plain language. This approach ensures that readers can grasp the essence of our work without getting overwhelmed by technical details. For example, we describe how the optimization framework aims to balance trading gains, liabilities, and associated costs dynamically over time to achieve a neutral Profit and Loss.
>
> To preserve mathematical accuracy and effectively engage readers who seek a deeper understanding, we have moved the detailed derivations supporting the final optimization goal to the Appendix B. This structure allows readers to access the technical depth they need without detracting from the narrative flow of the main text.

---

> ### Author Response · Authors · 2025-01-16
> **Addressing Confusion in Deep Hedging Formulation and Improving Readability**
>
> We appreciate the reviewer's detailed feedback highlighting areas of confusion and potential improvement in Section 4 in the original manuscript. We acknowledge the need for a clearer presentation of the methodology, focusing on high-level insights while relegating intricate mathematical details and intermediate notations to the appendix. These revisions aim to address the reviewer's feedback comprehensively.
>
> In the revised manuscript, Sections 3 and 4 have been extensively streamlined to emphasize the core innovations of our approach and ensure accessibility for a broader audience. The previous version included intermediate variables such as $h$, $A$, $M$, and $H$, which, while important for technical derivations, added unnecessary complexity to the main text. These variables have been removed from Sections 4 and are now detailed in Appendix D, along with the full derivations. This restructuring allows the main text to focus on the conceptual understanding of our methodology.
>
> Section 4 focuses on the design and innovations of our proposed framework, including the linearized training procedure and its integration with financial principles like the Black-Scholes Delta anchor. The linearized training procedure is explained conceptually, emphasizing how it simplifies the optimization landscape, enhances training stability, and accelerates convergence. The text now avoids detailed mathematical expressions in favor of clear explanations, highlighting the practical advantages and relevance of these innovations to hedging tasks.

---

### Review · Reviewer_CAa7 · 2024-12-16

**Summary Of Contributions:**

1) Authors have presented a method that combines a deep-learning based neural network and traditional black-scholes delta to come up with an improved achor hedge strategy in financial markets.

Helpful financial markets background: Hedging positions are usually held by traders to safeguard their primary investments by taking greater risks , e.g : buying put options as hedge in case the primary asset they invested in goes down or selling call options to if they want greater profits etc.  Every option / hedging position has a strike price, expiration time and associated volatility of the underlying security which determines the price. At expiration time if the underlying asset falls in-the-money the option can be exercised to get the payoff. If underlying asset price is out-of-money then the option is worthless and the initial payment to buy the option is lost. In the case of selling options it is possible the trader takes much more risk than just losing initial cost as they'll be forced to buy / sell the underlying asset. For more details see : https://www.investopedia.com/options-basics-tutorial-4583012


2) Authors have provided all the required details about financial markets for a machine learning practitioner to understand the hedging problem. In this paper the primary goal is not to maximize the total portfolio value but rather come up with hedging strategy that reduces the impact of adverse market effects. The problem is formulated as predicting the series of trades to be performed on the underlying asset till the expiry of the option such that the total cash injected into the position to make the trade safe as per a risk measure is minimized. At each time step the black scholes delta can be used to determine the probability that the option is in-the-money.  The proposed neural network augments the black scholes delta by providing upper and lower adjustments to the delta making it a band. The neural net takes as input the following features : log-moneyness, expiration time and volatility as these are the most crucial factors determining the price of an option. The proposed network produces 2 outputs representing the higher and lower corrections needed to black scholes delta.

3) Experiments are performed using both simulated and real price trajectories and the results are presented as profit-loss distributions (pnl) where it's observed that the proposed strategy is more robust in high transaction-cost situations and in situations with high volatility.

**Audience:**

Yes

**Claims And Evidence:**

Yes

**Requested Changes:**

It'll make the paper stronger if all the analysis under the section "Analysis of Approximation Quality in Hedging Model" would go into appendix making it much easier for readers to grasp the core message of this paper.

**Strengths And Weaknesses:**

Strength:  Authors have done a good job at formulating the hedging objective problem with detailing all the prerequisites needed to understand for a reader with no financial background.

Weakness:  Real markets might experience higher volatilities than what's tested in the paper with the fixed dataset so it's unclear how this strategy would perform in real markets.

---

> ### Author Response · Authors · 2025-01-16
> **Addressing Volatility Variations in Real Markets**
>
> We appreciate the reviewer's observation regarding the potential for higher volatilities in real markets and the importance of understanding how our strategy performs under such conditions. In the revised manuscript, we have taken several steps to address this concern and demonstrate the robustness of our proposed methodology.
>
> First, our experimental design explicitly incorporates out-of-distribution testing scenarios to evaluate the generalizability of our approach to higher levels of volatility beyond the training data. While the training dataset is generated using a fixed volatility of $\sigma = 0.1$, the testing phase introduces significantly higher volatilities, including $\sigma \in \{0.2, 0.3, 0.4\}$. These out-of-distribution tests provide a rigorous evaluation of the model's ability to adapt to market conditions characterized by greater uncertainty and more significant price fluctuations. The results from these experiments, detailed in Section 5.3, demonstrate that our DHLNN framework maintains robust performance, with stable and concentrated PNL distributions and minimal degradation in risk metrics, even as volatility increases.
>
> Second, real-world datasets were included in our experiments to complement the synthetic data and validate the practical applicability of our approach. The real market data tests involve complex financial instruments such as Lookback options, which are particularly sensitive to historical price extremes. The results from these experiments, presented in Section 5.4, further confirm the robustness and effectiveness of our approach in dynamic and volatile market environments. The combination of synthetic and real market data provides a comprehensive evaluation framework, bridging the gap between controlled experimental conditions and real-world applications.
>
> Lastly, our methodology incorporates a linearized training procedure designed to enhance stability and efficiency, particularly under volatile market conditions. By decoupling gradient computation from parameter updates and leveraging periodic fixed-gradient optimization, our approach simplifies the optimization landscape, reduces sensitivity to noisy data, and enhances generalization to extreme scenarios. These methodological advancements ensure that the DHLNN framework is not only computationally efficient but also robust against market frictions and sudden shifts in volatility.
>
> We recognize that real markets may experience even higher or more abrupt volatility changes than those tested in this study. To address such scenarios further, future work could involve extreme market events into the simulation framework. Nonetheless, the current results strongly suggest that the proposed DHLNN framework is well-suited to handle a wide range of market conditions, including those characterized by elevated volatility.

---

> ### Author Response · Authors · 2025-01-16
> **Streamlining the Presentation of Analysis for Clarity**
>
> We appreciate the reviewer's suggestion, we have moved the detailed analysis under the section ``Analysis of Approximation Quality in Hedging Model'' to the appendix. We also enhance the readability of the paper by reorganize the formulations, allowing readers to focus on the core contributions while retaining the detailed theoretical insights for those interested in the technical underpinnings of our approach.

---

### Decision · Action_Editor_uYJ5 · 2025-04-07

**Recommendation:** Reject

**Comment:**

The paper develops a general framework, i.e., Deep Hedging with Linearized-objective Neural Network (DHLNN),  which enhances the training procedure of deep learning models, and discusses its applications in finance domains. All reviewers agree that the paper aims to address an important problem, but one reviewer has significant concern regarding the presentation and the experiments. I read the paper and agree with the assessment.

The following changes needs to be made in order for the paper to be ready for publication:
1. Improve presentation by adding more explanation and references on key financial concepts;
2. Improve the experiment design: please discuss the choice of hyperparameter, Monte Carlo settings, and reproducibility; please also add more baselines using traditional methods (such as Delta hedging, quadratic hedging) as well as RNN/Transformer-based models.

**Audience:**

The paper has limited interest from the TMLR audience since it focuses a specific task, i.e., risk management, in financial applications.

**Claims And Evidence:**

The claims in the submissions are not supported by convincing evidence. The presentation can be improved by adding adequate explanation and references on key financial concepts. The experimental design (dataset choices etc) are not fully justified.  There is a limited number of baseline methods in the experiments.

**Resubmission Of Major Revision:**

The authors may consider submitting a major revision at a later time.